# An avian cortical circuit for chunking tutor song syllables into simple vocal-motor units

Emily L. Mackevicius [1], Michael T. L. Happ[1] & Michale S. Fee [1✉]

How are brain circuits constructed to achieve complex goals? The brains of young songbirds develop motor circuits that achieve the goal of imitating a specific tutor song to which they are exposed. Here, we set out to examine how song-generating circuits may be influenced early in song learning by a cortical region (Nlf) at the interface between auditory and motor systems. Single-unit recordings reveal that, during juvenile babbling, Nlf neurons burst at syllable onsets, with some neurons exhibiting selectivity for particular emerging syllable types. When juvenile birds listen to their tutor, Nlf neurons are also activated at tutor syllable onsets, and are often selective for particular syllable types. We examine a simple computational model in which tutor exposure imprints the correct number of syllable patterns as ensembles in an interconnected Nlf network. These ensembles are then reactivated during singing to train a set of syllable sequences in the motor network.

[1] Department of Brain and Cognitive Sciences, McGovern Institute for Brain Research, Massachusetts Institute of Technology, Cambridge, MA 02139, USA. ✉email: fee@mit.edu

Unlike motor circuits for innate behaviors, which are built by genetically specified developmental programs, motor circuits for complex learned behaviors must be built by a combination of genes and experience. Many of our most complex and expressive behavioral repertoires, from speech to music to cooking to sports, are learned through observing others, and through practice of simpler elements. For example, in the process of mastering a musical piece, musicians repeatedly practice very short musical elements. The process of learning often involves breaking a complex sequence of actions down into simple discrete pieces which are practiced individually and then assembled later. Such behavioral elements, or movement chunks, can be flexibly composed in the brain to produce complex motor sequences[1–3]. This powerful strategy for generating new behaviors supports the creation of a rich variety of actions, even from relatively few primitive building blocks. We were interested in how the brain makes building blocks for a new behavioral repertoire.

Songbirds are an excellent model system to address this question. Songbirds learn their vocalizations by imitating songs of tutors they heard as juveniles. They learn in two stages: a sensory stage where they memorize the tutor song, and a sensorimotor stage where they practice vocalizing[4,5]. The memory of the tutor song, called the song template, is sufficient to guide imitation[5], even if restricted to a single 2-h window of tutor exposure[6]. Song learning exhibits convergent behavioral, circuit-level, and genetic parallels with human speech learning[7–17], and also has parallels with other forms of mammalian motor learning[18].

Like other complex motor behaviors such as speech, songs are divided into discrete simpler chunks, syllables[19]. After initial random babbling, a repeatable protosyllable emerges, which then differentiates into multiple daughter syllables[4]. As the song matures, syllables undergo a process of gradual refinement. The neural mechanisms of adult song production and refinement are fairly well understood, and rely, like mammalian motor learning, on a network of cortical, basal ganglia, and thalamic brain areas[13,16,18]. Together, this network of brain areas is thought to explore new song variations[20–23]; evaluate which variations sound good[24–27]; employ reinforcement learning to bias future song towards variants that sound better[28–30]; and ultimately execute precisely timed neural sequences to generate each song syllable[31–39].

What neural signals might help build the sequences that precisely control adult song syllables? Song timing is thought to be controlled by HVC[31,32,35,36], where neurons generate brief bursts in precisely timed sequences that span every moment in the song[38,39]. These precise sequences appear to emerge gradually over development, first with the growth of protosequences from syllable onsets, then the splitting of protosequences into daughter sequences[40]. Several existing computational models of HVC development rely on a population of syllable-onset-related training neurons that seed developing chains in HVC[40,41]. In these models, HVC receives brief bursts of external input, and the HVC network organizes into chains of sequentially active neurons that are triggered by inputs to training/seed neurons.

Based on a large body of previous work, we hypothesized that seed neurons in HVC may be driven by a signal from the auditory system, and that this could train the HVC network to form syllable-sized chains[42]. At a computational level, we view song imitation as learning a generative model of the tutor song. Exposure to a tutor song could imprint the desired collection of syllable-sized chunks in the auditory system[43,44], which through interaction with HVC could then create an appropriate latent representation of song timing in the form of HVC sequences. More specifically, we proposed that auditory cortex may directly influence sequence formation in HVC by appropriately activating seed neurons during development. Several experimental results are consistent with this view: tutor exposure produces rapid overnight changes in the song motor system, including dramatic alterations of song features[4], spontaneous activity[45], and spine growth and stabilization[46]. Despite the apparent importance of auditory inputs to HVC during learning, these inputs have little role in adult singing, and indeed are gated off in adult birds[47,48].

Several lines of evidence led us to focus on nucleus interface (NIf), a higher-order sensorimotor cortical area that projects to HVC and has been implicated in song learning. Like mammalian association cortex, NIf is a central part of an interconnected network of auditory and motor cortical areas[49–54] (Fig. 1a).

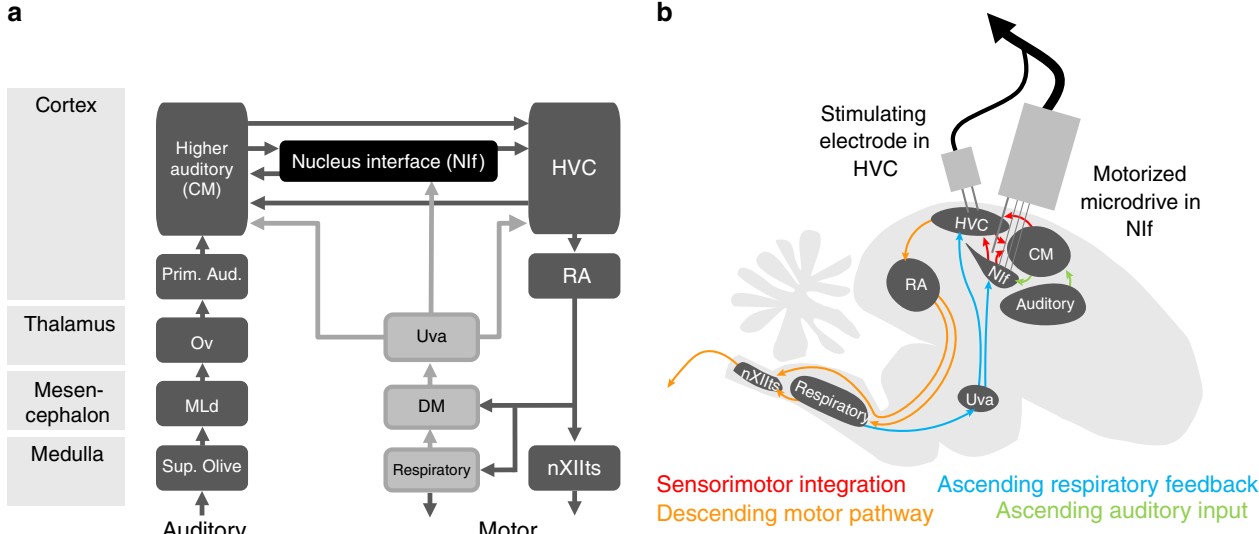

**Fig. 1 Recording at the interface between sensory and motor cortical circuits in juvenile birds during song learning. a** Schematic showing interactions of the ascending auditory pathway and descending motor pathway in the songbird brain. Like association cortex in mammals, NIf is densely connected with both sensory and motor areas. **b** Schematic of the songbird brain showing interconnected sensory and motor nuclei. A motorized microdrive was implanted with electrodes in NIf, to enable recordings in freely behaving birds. In addition, a stimulating electrode was implanted in the premotor cortical nucleus HVC to antidromically identify NIf_HVC neurons.

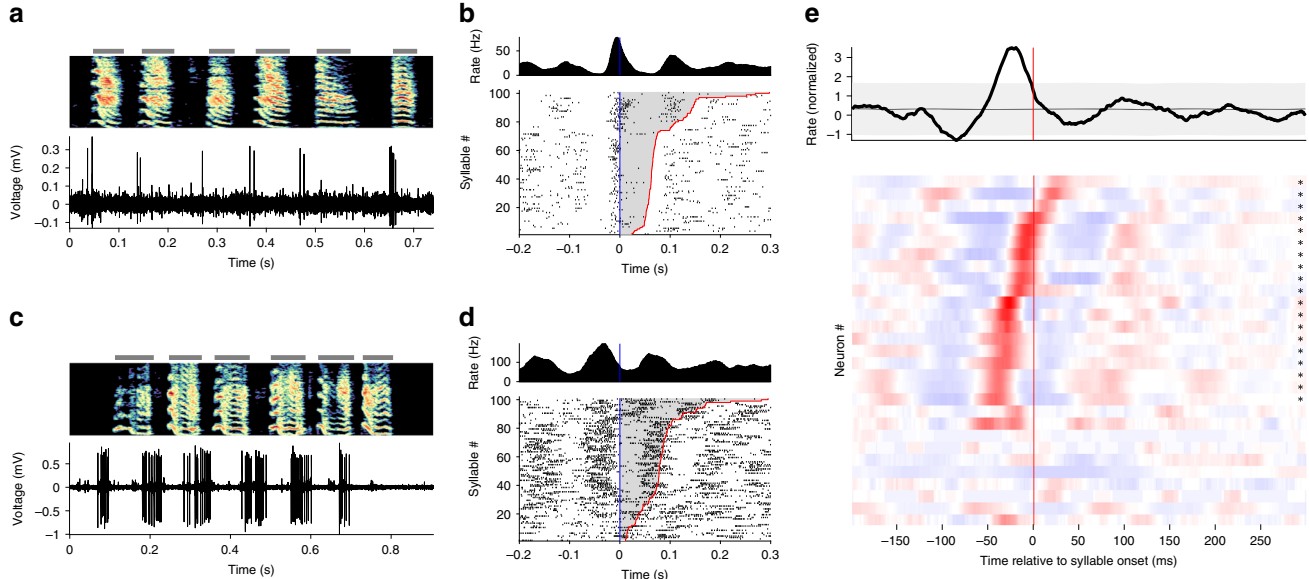

**Fig. 2 Onset-related activity in NIf during singing in juvenile birds. a–d** Examples of NIf$_{HVC}$ single-neuron activity during singing in juvenile birds. **a** Song spectrogram (top), and single-unit recording (bottom). Gray bars indicate the extent of each syllable. Here and elsewhere, the vertical axis of the spectrogram represents frequencies ranging from 0.5 to 8 kHz. Note the neuron tends to burst at syllable onsets. **b** Onset-aligned raster and PSTH for neuron at left, sorted by syllable duration. Syllable onset is marked in blue, offset in red, with the duration of each syllable shaded gray. **c**, **d** Same as **a** and **b** for another putative NIf$_{HVC}$ recorded neuron during singing. Note that this neuron has a longer premotor latency, longer burst durations, and higher firing rates. **e** Summary of all NIf$_{HVC}$ single-unit recordings during singing. The plot at top shows the mean normalized firing rate across all neurons, with gray region showing the extremes of a circularly shifted control ($p < 0.01$, two-sided comparison). Plotted below is the syllable onset-aligned PSTH for each individual NIf$_{HVC}$ neuron; each row corresponds to the PSTH of a different neuron. PSTHs are normalized by subtracting the mean and dividing by the standard deviation of the PSTH; red indicates values above the mean, blue below the mean. Neurons with a significant PSTH peak are marked by an asterisk at right ($p < 0.05$, one-sided comparison to circularly shifted control, Bonferroni corrected for 29 comparisons). Significant neurons are sorted by the latency of their peak firing rate, and non-significant neurons are shown at bottom.

Specifically, it has been suggested that NIf is analogous to Spt, a region in the human speech processing circuit located at the interface of auditory and motor functions[53,55], where neural activity is locked to onsets of words and phrases[56]. NIf appears to play a key role during vocal development: lesions of NIf have minimal effect on adult zebra finch song[57], but inactivation of NIf in young juvenile birds causes loss of emerging spectral and temporal song structure[58], and inactivating NIf while a juvenile bird is being tutored interferes with tutor imitation[59]. Furthermore, NIf neurons that project to HVC have been observed to burst at syllable onset[60], and NIf lesions affect syllable ordering[61], consistent with a role for NIf in chunking song into syllables.

To test our hypothesis that NIf provides a training input to seed neurons in HVC, we record from neurons in NIf in freely behaving juvenile zebra finches (Fig. 1b) while they sing and while they listen to a tutor song. We find that NIf exhibits bursts of activity near syllable onsets in juvenile birds, as has previously been reported[60]; we additionally find that NIf neurons exhibit syllable-specific activity, tending to burst more at the onsets of some syllables than others. In addition, our recordings during tutoring reveal that NIf also fires at syllable onsets, chunking the song into syllables in both singing and listening contexts. Thus, activity in NIf, or upstream of NIf, may serve to align emerging auditory and motor representations of song, allowing a stable reference frame for song learning. We develop a mechanistic neural network model, in which synaptic learning rules lead the NIf network to form a distinct ensemble for each tutor syllable. In a combined NIf/HVC model, these NIf ensembles are sufficient to train downstream premotor sequences in HVC. Finally, in light of recent optogenetics results[62], we model how NIf may play a role specifying the durations of song syllables.

## Results

**Singing-related activity of NIf neurons in juvenile birds**. We set out to characterize the neural activity patterns of NIf neurons in singing juvenile birds (89 single-unit recordings from 13 birds ages 44–92 dph, see "Methods" section). We recorded 29 antidromically identified neurons, as well as 60 neurons that were within the borders of NIf as determined by antidromic hash, but which did not meet our criteria for identified projectors. Some of these non-identified neurons could be HVC projectors that were not stimulated, while others could be local NIf interneurons. We found that many identified HVC-projecting NIf (NIf$_{HVC}$) neurons generated bursts that were strongly locked to syllable onsets (Fig. 2a–d). A population average over all projection neurons revealed a robust peak in the average firing rate 19 ms prior to syllable onset (Fig. 2e, top; $p < 2 \times 10^{-6}$, see "Methods" section). A more detailed analysis reveals that 66% (19/29) of the projection neurons exhibited a significant firing rate peak within a window 50 ms before to +25 ms after syllable onsets ($p < 0.05$, with Bonferroni correction for 29 comparisons). Some neurons produced narrow bursts aligned immediately prior to syllable onsets (Fig. 2a, b, and e), and others produced wider bursts with longer premotor latencies (Fig. 2c–e), and still others burst shortly after syllable onset (Fig. 2e). This range of latencies in peak firing rates relative to syllable onset extended from 41 ms before syllable onset to 25 ms after (Fig. 2e). Across the song-locked NIf$_{HVC}$ neurons, the median latency of the peak was 16 ms prior to syllable onset. Note that most neurons that exhibited a significant peak at syllable onset also exhibited flanking regions of lower-than average firing rates, a pattern apparent in the population-average (Fig. 2e, top).

Turning now to the population of neurons not identified as HVC projectors, the average PSTH of these neurons during singing revealed a prominent peak at 15 ms after syllable onset

($p < 2 \times 10^{-6}$, Supplementary Fig. 2). A detailed analysis reveals that 35/60 individual neurons exhibited a significant peak in firing rate in the −100 to +25 ms range. Of these, 19 neurons had a firing rate peak in a narrow window 0–25 ms after syllable onset, while 16 neurons had a broader range of peak times prior to syllable onset, similar to the identified projection neurons. Overall, this somewhat bimodal distribution of properties among non-identified neurons might be consistent with the idea that this population comprises two distinct neuron types. The latter of these might correspond to HVC-projectors that were not stimulated in our antidromic identification protocol.

Earlier models of sequence formation in HVC have proposed that external inputs at subsong syllable onsets drive seed neurons in HVC that initiate the growth and splitting of synaptic chains within the HVC network[40]. Indeed, during subsong, many HVC projection neurons fire reliably at syllable onset[40]. To test the idea that such external input arises from NIf, we recorded six NIf neurons (including four NIf$_{HVC}$ neurons) during subsong. We found NIf neurons burst at syllable onset even at this early stage of song development (Fig. 3a–c).

Consistent with the possibility that NIf neurons function to activate HVC chains at syllable onsets, we found that the distribution of latencies of NIf neurons is much earlier and tighter than that of HVC neurons. The latencies of individual NIf$_{HVC}$ neurons are clustered prior to syllable onsets at all stages of song learning, with 80%, 79%, and 83% of bursts occurring prior to syllable onset in subsong, protosyllable, and multisyllable stages, respectively (Fig. 3c). In contrast, the distribution of HVC burst latencies appears to change over development in a way consistent with chains growing from syllable onsets[40], such that in adult birds HVC bursts occur fairly uniformly throughout the song motif[38,39]. These results are consistent with a model in which HVC sequences grow from syllable onsets, triggered by inputs from NIf (Fig. 3e). We note that, while the peaks of the PSTHs of individual NIf neurons occur primarily at syllable onset, individual NIf neurons have also been observed to fire at moments other than syllable onset, for example, at transition points within multi-part syllables[60]. This observation is consistent with the possibility that some syllables may be formed by combining multiple protosyllables[40].

We wanted to ensure that the observed narrow distribution of NIf latencies was not an artifact of biased spatial sampling in NIf. NIf is an elongated nucleus that sits along the mesopallial lamina (Supplementary Fig. 3, inset). Recent work has shown that projections to HVC display non-uniform topology, and lesions to different parts of HVC may result in different types of song deficits[63–65]. Using post-hoc histology, it was possible to estimate the anatomical position within NIf of some of our recording sites, and we found that our recordings were sampled from the entire extent of NIf. Notably, no significant correlation between response latency and the anatomical position of the recording site was observed ($R^2 = 0.15$, Supplementary Fig. 3).

We were curious whether NIf onset-related activity during singing is selective for specific emerging syllable types, or whether the activity is equivalent for all syllable types. In an intermediate stage of song learning, multiple syllable types emerge from a common protosyllable, and multiple HVC sequences appear to emerge from a common protosequence[4,40]. We find that some NIf$_{HVC}$ neurons burst preferentially at the onsets of certain syllable types (Fig. 4). Of the song-locked putative HVC projection neurons we recorded in juveniles with multiple syllable types, most had significantly different firing rates for different syllables (6/8 pass ANOVA at 0.05 significance level with Bonferroni correction). These results are consistent with a potential role for NIf in shaping the emergence of syllable-specific sequences in HVC.

**NIf activity during tutoring**. It has previously been proposed that auditory experience of the tutor song can directly drive the formation of song sequences in HVC[42,66]. To investigate the potential role of NIf in mediating this hypothesized process, we recorded from NIf in juvenile birds during early tutor exposure (103 single-unit recordings from 14 birds between the ages of 41 and 67, including 29 putative HVC-projectors). Approximately half of these neurons produced clear bursts of activity during presentation of the tutor song (Fig. 5). The population average syllable-aligned PSTH exhibited a significant peak near syllable onset with a latency of 14 ms post onset (Fig. 5g; $p < 2 \times 10^{-6}$). The onset-related peak was observed separately in both the population of NIf$_{HVC}$ projection neurons (peak latency 2 ms; $p < 1 \times 10^{-4}$) and in the population of non-identified neurons (peak latency 15 ms; $p < 2 \times 10^{-6}$).

A more detailed analysis of individual neurons revealed that 51 of all 103 neurons (including 8 of 29 projection neurons) exhibited significantly elevated firing rates in a window between 0 and 25 ms after tutor syllable onset (Fig. 5h, i; $p < 0.05$, Bonferroni corrected). Among individual neurons with significant responses, the median latency was 14 ms after syllable onset. Interestingly, we also observed a small subset of neurons that produced a dip in firing rate in the 0–25 ms window after syllable onset (6/103 neurons, including 1/29 projectors; $p < 0.05$, Bonferroni corrected for 103 comparisons). For these neurons, the average firing rate change in this window was −7 Hz, compared to the +22 Hz firing rate change for the 51 neurons exhibiting an increase. Thus, as a population, the predominant modulation in NIf was a brief increase in firing rate at syllable onsets. As in the singing data, no significant correlation was observed between latency and the position of the recording site. Of the neurons we were able to record during both singing and tutoring, some were exclusively singing-locked (12/24 neurons, including 7/10 projectors), some were both singing-locked and tutor-locked (8/24 neurons, including 1/10 projectors), a few were exclusively tutor-locked (2/24 neurons, including 1/10 projectors), and the remaining 2/24 neurons did not show changes in firing rate related to singing or tutoring. Of the eight neurons that were locked at syllable onsets during both singing and tutoring, we observed a range of singing latencies from −6 to +18 ms, with a mean of 9.6 ms, and a range of tutoring latencies from +6 to +45 ms, with a mean of +20.3 ms. For the one HVC-projecting NIf neuron, the singing latency was −2 ms, and the tutoring latency was +26 ms.

We wondered whether NIf is active at syllable offsets as well as syllable onsets. For most syllables in the tutor song, it is difficult to decouple syllable offset responses from onset responses, because the offset of most syllables is followed by the onset of the next syllable in the song. Therefore, to analyze syllable offsets, we aligned NIf activity to the offsets of the final syllable in the tutor song bout. As a population, NIf neurons exhibited no significant modulation (either increase or decrease) following last syllable offset (Supplementary Fig. 4A). Only 2/103 neurons (neither of which were projectors) exhibited a significant increase in spiking activity in a 0–25 ms window following last-syllable offset ($p < 0.05$, Bonferroni correction for 103 comparisons, Supplementary Fig. 4B, C). Interestingly, both of these neurons also exhibited robust syllable-onset responses.

The hypothesis that NIf may translate the tutor song into an appropriate collection of HVC sequences suggests that NIf activity during tutoring may be syllable selective. Notably, of the putative projection neurons that had significantly elevated responses to tutor syllable onsets, most (6 of 8 neurons) also had different firing rates for different syllable types (Fig. 6; ANOVA at $p < 0.05$ with Bonferroni correction). For these selective

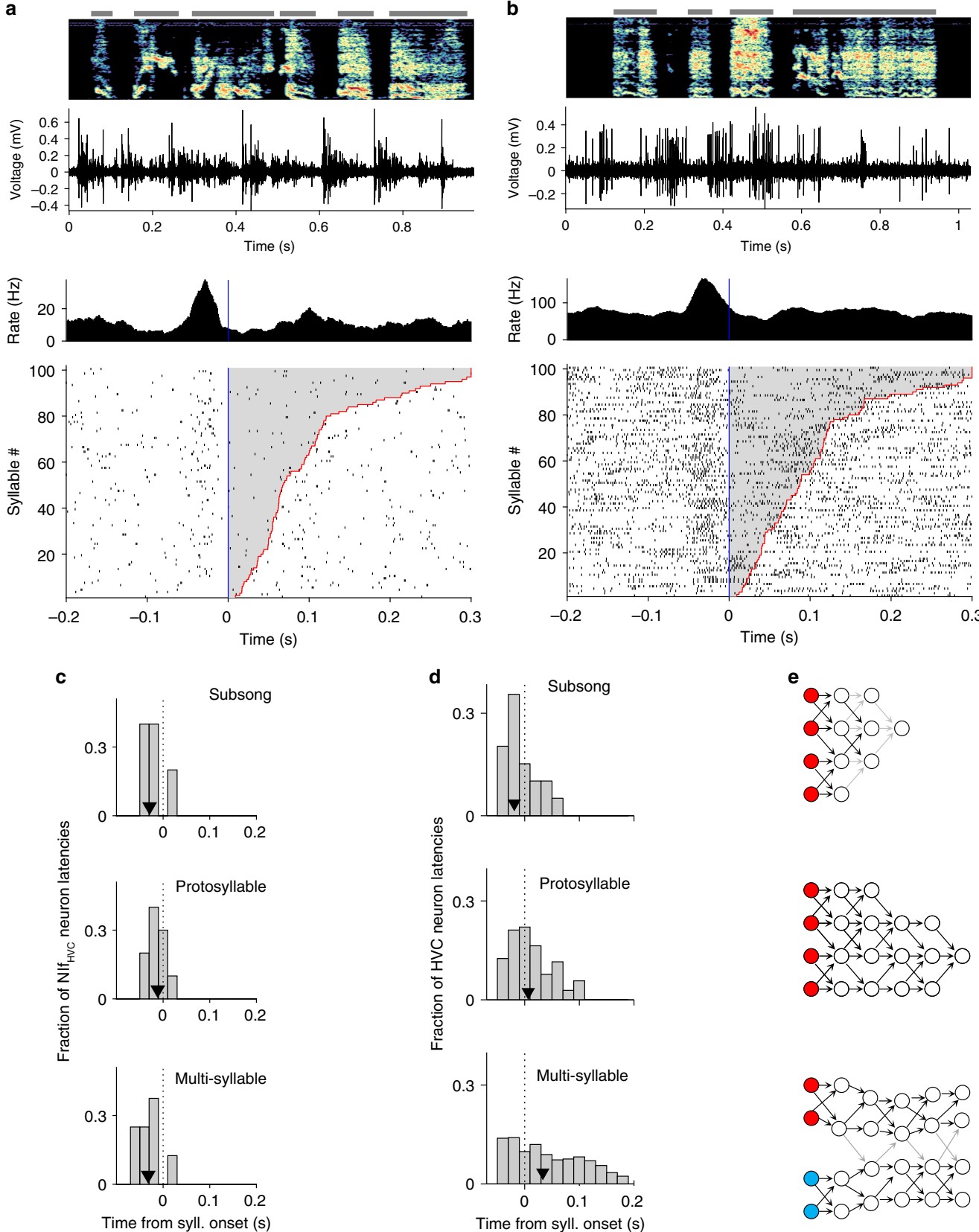

neurons, firing rate variances across syllable types were 13 times larger than within syllable types (*F*-test statistic was on average 13). These findings are consistent with the idea that during tutoring, NIf exhibits strongly syllable-related and syllable-specific activity.

**Mechanistic model and discussion.** The recordings described above hint at how the brain may translate an auditory memory of the tutor song into motor commands to generate a precise imitation. Here, we formalize these ideas into a broader computational framework that includes both NIf and the downstream

**Fig. 3 Distributions of response latencies relative to syllable onset at different stages of vocal development. a** An example putative $NIf_{HVC}$ neuron recorded during subsong (49 dph), the earliest stage of vocal development. **b** Another example of a NIf neuron recorded during subsong (47 dph). Note the premotor latencies relative to syllable onset. **c** Histogram of latency relative to syllable onset for song-locked putative $NIf_{HVC}$ neurons at subsong (50 dph or younger), protosyllable (51–59 dph), and multi-syllable (60 dph or older) stages. Latency is defined as time of the peak in the syllable-onset-aligned PSTH. Triangles indicate median. **d** For comparison, histogram of latency relative to syllable onset for HVC projection neurons across development, reproduced from ref. [40]. Here, song stages are defined based on syllable duration distributions, with median and interquartile range 48 ± 4 dph for subsong; 58 ± 10 dph for protosyllable song and 62 ± 12 dph for multi-syllable stage ($n = 19$, 104, and 814 neurons, respectively). **e** Schematic of possible HVC network connectivity at each stage of development[40], consistent with latencies observed in HVC and NIf. Neurons receiving external input are marked by filled circles, with different colors representing syllable-specific inputs. Connections between neurons are marked by arrows, where darker arrows represent stronger connections.

premotor area HVC. We build on previous models of HVC[40] by replacing a hard-coded training input with a model of NIf that learns the structure of the tutor song and autonomously generates and transmits a training signal to HVC. First, we present a computational model of NIf (Fig. 7a, b): The key idea is that exposure to a tutoring input imprints a pattern of connectivity within NIf that defines an ensemble of neurons for each syllable in the tutor song. These ensembles are then re-activated later during singing. The activity of the model during the tutoring and singing stages captures features of our NIf recordings. We next combine the NIf model with a model of HVC, and describe how the NIf network learns an outline of song structure in the auditory domain and translates this structure to the model HVC network to guide learning in the motor domain. Finally, we use the combined NIf/HVC model to demonstrate a potential mechanism by which NIf may control the duration of syllable sequences in HVC, consistent with recent optogenetics findings that stimulation of NIf at different rhythms can imprint different syllable durations[62]. Note that this captures only the very earliest stages of learning, and does yet not include a mechanism to read out song errors, which is thought to occur separately[18,26].

We set out to design a model NIf network that generates a distinct neural ensemble for each syllable in the tutor song, then reactivates those same ensembles autonomously during singing. To achieve this, we took inspiration from existing models of neural ensemble formation[67,68], and used learning rules that both associate neurons together in clusters, and also compete clusters against each other (details below). Inputs to the network consist of two classes of syllable-related inputs (Fig. 7a). The first is a syllable-specific auditory input that is only active during tutoring and consists of a different random pattern of activity for each syllable in the tutor song. The second input is a syllable onset signal active during both tutoring and singing. During tutoring, this signal, derived elsewhere in the auditory system, potentiates NIf at syllable onset. During singing, this onset signal corresponds to an efference copy of preparatory motor commands originating in brain areas responsible for the initiation of early babbling syllables. The role of this efference copy input is to reactivate syllable-specific NIf populations during singing. Together, these two inputs lead to the formation of syllable-specific ensembles during tutoring (Fig. 7d, f, g) and to the reactivation of these ensembles during singing (Fig. 7e, h), when they serve as seed inputs to drive chain formation in HVC.

Before describing the model further, we briefly elaborate on evidence that syllable onset activity in NIf in early song vocalizations corresponds to an efference copy of motor commands originating elsewhere (Fig. 7a). One line of evidence arises from studies suggesting that subsong babbling is driven by a cortical nucleus lateral magnocellular nucleus of the nidopallium (LMAN), rather than the nucleus HVC that drives adult song. While both HVC and LMAN exhibit strong syllable onset activity in subsong, lesions of LMAN completely abolish these early vocalizations, while lesions of HVC have little effect[40,69]. LMAN is

thought to activate subsong syllable onsets through a pathway to RA and then to brainstem vocal and respiratory centers. An efference copy of these onset signals could reach NIf via a feedback pathway from midbrain vocalization and respiratory centers through the thalamic nucleus Uvaeformis (Uva)[70], a pathway thought to provide interhemispheric synchronization[71,72], coordinate respiration[34,73,74], as well as control the sequential generation of syllables in a song motif[75,76], or sequential structure within syllables[77]. These ideas are further supported by the recent observation that Uva exhibits strong syllable onset-related signals during singing in adult birds[78]. Notably, efference copy signals could also reach HVC directly from Uva, or from Uva through another higher-order auditory area (CM), or via Area X and A11[79], pathways that could also, in principle, play a role in sequence formation in HVC.

How do we construct a model such that, during the tutoring phase, auditory inputs give rise to the formation of syllable-specific ensembles? In the simplest case, auditory inputs for each syllable would activate non-overlapping populations of NIf neurons, which would then form independent ensembles for each syllable through Hebbian learning. In reality, auditory inputs for each syllable do overlap with each other, so the main challenge is to form independent ensembles in NIf in spite of overlapping inputs. There are many different ways to accomplish this goal[67,68]; in our implementation, we take inspiration from previous work demonstrating that anti-Hebbian learning can decorrelate input patterns[67,80]. Thus we initiate the tutoring phase with a brief episode of anti-Hebbian learning during which synapses between co-active neurons are made more negative. This ensures that auditory inputs for different syllables activate non-overlapping ensembles of neurons in the later tutoring stage. After the episode of anti-Hebbian learning, tutoring continues with a Hopfield-like Hebbian learning rule, which burns each non-overlapping population into the network as an independent, recurrently connected ensemble for each syllable (Fig. 7e). The full progression of network connectivity over time is shown in Supplementary Movie 1.

In our model, these syllable-specific ensembles are reactivated during singing as attractors of the NIf network by the rhythmically driven syllable onset signal. Because the onset signal has no syllable-specificity, the specific ensemble reactivated is determined by either noise or initial conditions. In our implementation, we have used an adaptation term (see "Methods" section) that prevents the same ensemble from being repeatedly activated on subsequent cycles of the onset signal. Thus, the model network cycles reliably through the stored syllable ensembles. While we did not set out to explicitly model how the syllable order is represented or controlled by the songbird brain, it is likely that weak, slow feed-forward connections between ensembles in NIf could, in principle, bias the song to reproduce syllable orderings presented during tutoring[81].

The result of the NIf model is a network that organizes into recurrently connected ensembles corresponding to the different

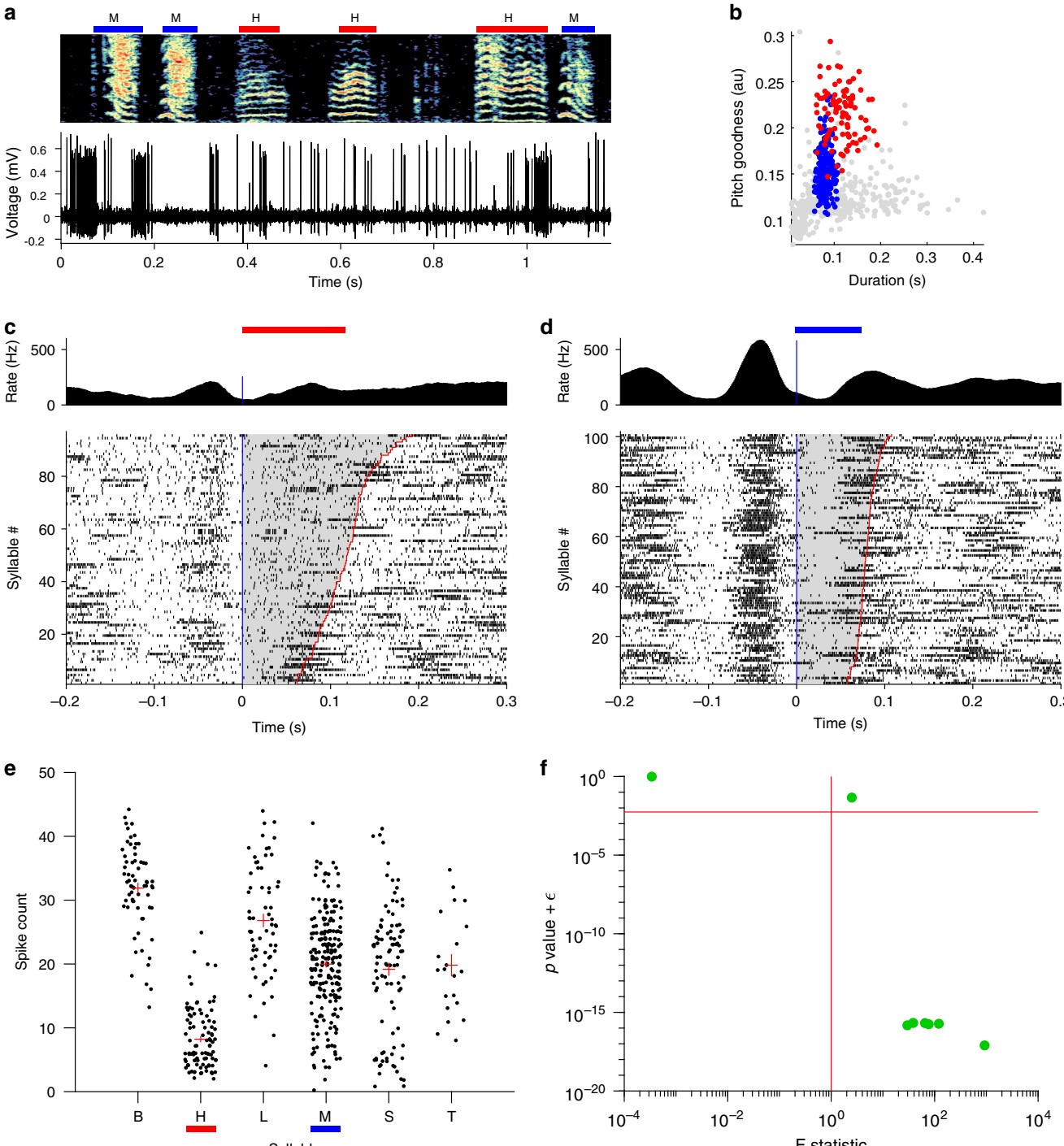

**Fig. 4 Syllable selectivity of NIf_HVC neuron activity during singing. a** Example of a NIf neuron recorded as a juvenile bird sang two different syllable types, marked H and M. **b** Scatter plot in acoustic feature space of all syllables the bird sang while this neuron was recorded, colored by syllable type. **c** Onset-aligned raster and PSTH for syllable H, marked by red bar. **d** Onset-aligned raster and PSTH for syllable M, marked by blue bar. Note that this neuron fires more before syllable M than syllable H (two-sample *t*-test, $p < 48.2e-34$). **e** Scatter plots showing spike counts of this neuron across six different syllables types. Spike counts were calculated in a window from 50 ms before syllable onset to 20 ms after syllable onset. Datapoints are jittered for ease of view. A one-way ANOVA was run to assess whether different syllable types had significantly different mean spike counts ($F = 83.6$; $p = 2 \times 10^{-65}$). Horizontal red lines indicate the mean spike count for each syllable type, and vertical red lines indicate the standard error of the mean. **f** Scatter plot showing one-way ANOVA *p*-values and *F*-statistics for all NIf_HVC neurons recorded in birds singing songs with multiple syllable types ($n = 8$ projection neurons from two birds). Horizontal red line shows Bonferroni corrected significance threshold for $p = 0.05$. Vertical red line indicates *F*-statistic of 1.

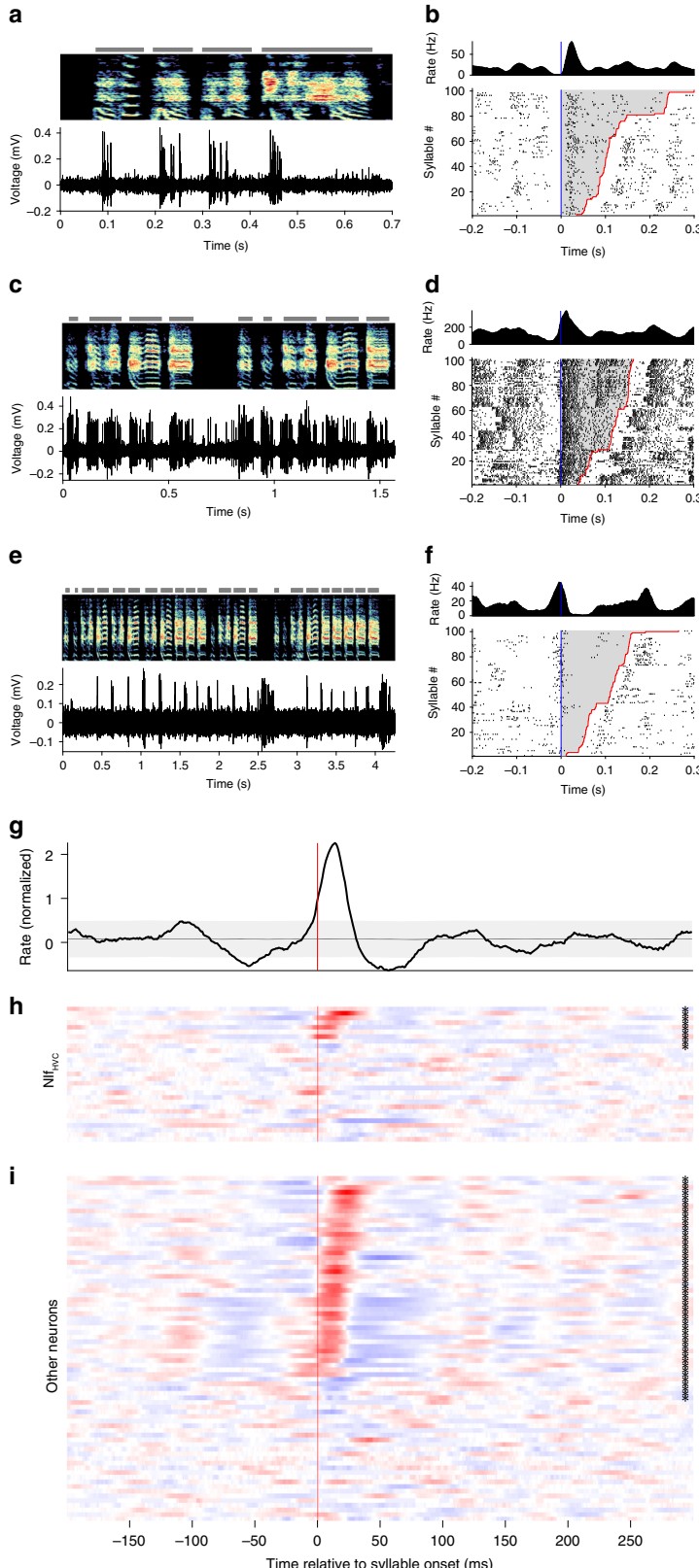

syllables presented during the tutoring phase. A numerical analysis of network performance reveals that the network can successfully form ensembles for each tutor syllable and robustly reproduce the correct number of independent syllables during singing. For example, when tutored with four syllables, the network both formed the correct number of syllable-specific ensembles during the tutoring phase and reproduced those syllables during the singing phase on 98/100 random initializations. The network was also largely successful when trained on three (81/100 successful runs) or five (79/100) syllables.

Previous models of HVC sequence formation incorporated syllable-specific training neurons as an input to HVC[40,42],

**Fig. 5 Neural activity in NIf during tutoring. a** Example NIf neuron recorded in a juvenile bird listening to tutor song. **b** Onset-aligned raster and PSTH for this neuron, sorted by tutor syllable duration. Note that this neuron fires at syllable onsets. **c–f** Two example NIf$_{HVC}$ neurons recorded during tutoring. **g** Summary of NIf activity during tutoring for all recorded neurons. The top panel shows the mean normalized firing rate across all of these neurons, with gray region showing the extremes of a circularly shifted control ($p < 0.01$, two-sided comparison). **h** Heatmap showing normalized onset-aligned PSTH for all recorded putative NIf$_{HVC}$ neurons. Neurons with significant firing rate increases at syllable onset are presented at top ($n = 8$) and sorted by the latency of their peak firing rate. Below these is shown the PSTH for one neuron with significant firing rate dips. Neurons with non-significant modulations are shown below. **i** Heatmap showing normalized onset-aligned PSTH for all recorded non-identified NIf neurons. Neurons with significant firing rate increases at syllable onset are presented at top ($n = 43$) and sorted by the latency of their peak firing rate. Below these is shown the PSTH for neurons with significant firing rate dips ($n = 5$). Neurons with non-significant modulations are shown below. For **h** and **i**, statistical significance was $p < 0.05$, one-sided, Bonferroni corrected for 103 comparisons.

consistent with the activity we subsequently observed in NIf (and report here). To further examine this hypothesized interaction, we connected the NIf model to the earlier model of HVC[40] and tested whether the replay of model NIf ensembles could train HVC to produce a unique sequence for each syllable in the tutor song. In the combined NIf/HVC model, activity in NIf was able to train HVC to assemble new sequences for each syllable in the tutor song (Fig. 8a). More specifically, in the combined model, HVC sequences grew in length and exhibited syllable-specific differentiation of activity, just as observed in HVC neurons recorded in developing birds. The combined model demonstrates explicitly how the formation of different ensembles in NIf can drive chain splitting in HVC such that different syllables in the song are represented by different chains in HVC.

Our combined NIf/HVC model also suggests a potential mechanism by which NIf may control the duration song syllables. Namely that the HVC network learns sequence durations that are strongly influenced by the period of activity imposed through NIf (Fig. 8b, c). This possibility was recently experimentally demonstrated by Zhao et al. in which patterned optogenetic stimulation of NIf in young birds was able to affect the durations of learned song syllables. For example, stimulating NIf at a fast 10 Hz rhythm led birds to sing shorter syllables[62], while stimulation of NIf with longer pulses (300 ms) at irregular intervals led to the formation of pathologically long syllables with a wide range of durations (from 150 ms to almost 1 s). We found that our combined NIf/HVC model precisely reproduces these findings: rhythmic NIf activation led to the formation of precisely timed syllables with a duration equal to the period of the stimulation, while irregular stimulation at long intervals led to highly variable durations of HVC network activity, corresponding to long syllables ranging from 150 ms up to a second in duration (Fig. 8b–d). Thus, in the combined NIf/HVC model, syllable durations appear to be controlled by a combination of NIf inputs and dynamics within the HVC network. NIf may activate HVC, which then 'reverberates' for a period of time until it is either reset by the next NIf input, or the reverberation in HVC dies out, as determined by its intrinsic recurrent excitation and inhibition.

Interestingly, the NIf network frequently displayed other behaviors besides strict imitation of the tutor patterns. These other behaviors are reminiscent of natural variations described in zebra finch song learning (Fig. 9). For example, while zebra finches often imitate their tutor, approximately half of birds generate song variations by deleting a syllable, improvising a new syllable, or duplicating an individual tutor syllable to produce two distinct syllable variants[25,82,83]. Similarly, during singing, our model network sometimes fails to activate a tutored ensemble (deletion, Fig. 9a), or sometimes activates a novel ensemble comprising neurons that were not activated together during tutoring (improvisation, Fig. 9b). Finally, during tutoring, the model sometimes forms two distinct ensembles activated by the same tutor syllable, both of which are then reactivated during singing, resulting in the duplication of that tutor syllable (Fig. 9c).

The relative prevalence of these variations depends on the choice of model parameters. For example, the adaptation time-constant during either tutoring or singing has a particularly prominent effect on these phenomena. Altogether, the model qualitatively recapitulates some surprising and unexplained aspects of vocal imitation in juvenile songbirds.

Our findings provide additional insight into potential mechanisms driving the mirroring of activity between sensory observation and motor action. The onset-related signals observed in NIf during both singing and tutoring give insights into how mirror neuron activity may arise[66,84,85]. In our view, the mirror activity in NIf is a signature of a sensory system that encodes a memory of a sensory event (the tutor song) subsequently used to train a motor system. The mirroring activity in the post-learning adult results from the vestigial neural architecture subserving the earlier learning process.

Together, our NIf recordings and modeling suggest potential neural mechanisms for breaking a tutor song into simple vocal-motor units that can be used to build a motor program that aligns with the auditory memory. By aligning activity to syllable onsets, NIf could chunk songs into syllables, each of which is accessible as a discrete unit during later motor learning and production. This view is consistent with the observation that NIf is necessary in some birds for flexible syllable sequencing[61]. In our model, simple neural plasticity rules produce an appropriate training input that reflects the structure of a tutor song. Building a motor program in this way, shaped by input from a sensory system, could create the precise alignment between a sensory memory and motor representations necessary for subsequent motor learning and evaluation. In the broadest view, this work elucidates neural mechanisms by which the brain, through sensory experience of the environment, can build motor programs appropriate to interact with that environment.

## Methods

**Subjects**. Electrophysiological recordings were carried out in 20 juvenile zebra finches (*Taeniopygia guttata*) from the MIT zebra finch breeding facility (Cambridge, MA). Birds were implanted with motorized microdrives between 37 and 42 days post hatch. We injected tracer (fluorescently labeled Dextran or Cholera-atoxin) in HVC to visualize NIf in posthoc histology. After recovery from surgery, birds were tutored using playback from a speaker with an adult bird present. Some birds (isolate birds) were raised by female birds, so this tutoring was their only exposure to any tutor song. Other birds were raised by both parents, and we played their own father's song during tutoring. We did not observe a significant difference between these groups of birds, so we combined the two groups in our analyses (17/21 song-locked in isolate birds with latency $-21 \pm 18$ ms relative to syllable onset; 37/68 song-locked in non-isolate birds with latency $-1 \pm 27$ ms relative to syllable onset; 27/48 tutor-locked in isolate birds with latency $15 \pm 7$ ms relative to syllable onset; 24/55 tutor-locked in non-isolate birds with latency $10 \pm 29$ ms relative to syllable onset). Once the birds started to sing, we also recorded during singing, and continued recording until either recording quality degraded or the bird's song developed a stable motif. Animal care and experiments were carried out in accordance with NIH guidelines, and reviewed and approved by the Massachusetts Institute of Technology Committee on Animal Care.

**Neural recordings in freely behaving zebra finches**. We recorded 168 single units in 20 juvenile birds during tutoring and singing. The electrophysiological methods that were used are described in detail in previous publications[86,87]. Some units were antidromically identified as HVC-projectors using a stimulating

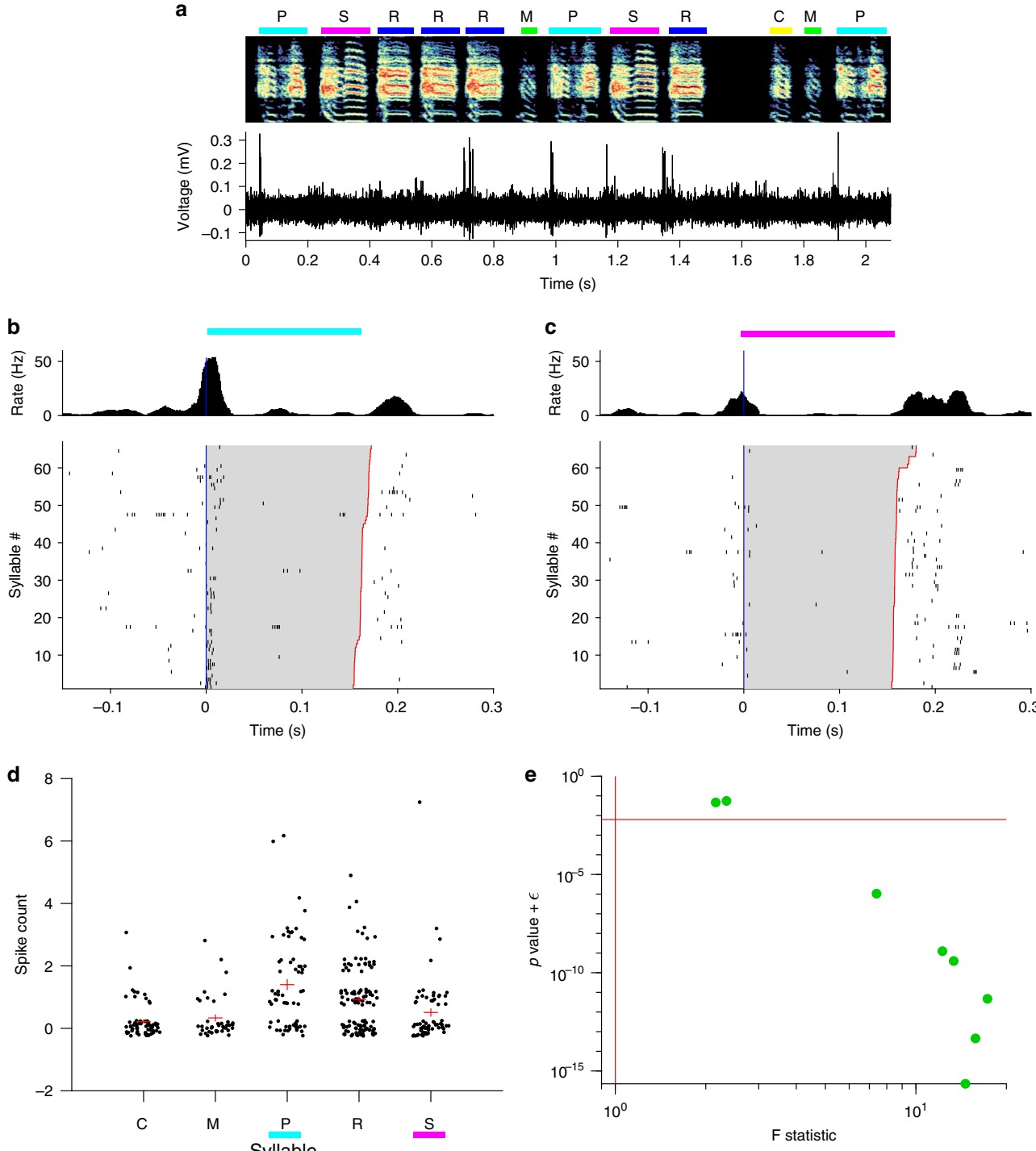

**Fig. 6 Syllable selectivity of NIf_{HVC} neurons during tutoring. a** Example of a NIf neuron recorded as a juvenile listened to different tutor syllable types, each marked by a different color. **b** Onset-aligned raster and PSTH for syllable P. **c** Onset-aligned raster and PSTH for syllable S. Note that this neuron fires more before syllable P than syllable S (two-sample $t$-test $p < 9.5e−05$). **d** Scatter plots showing spike counts of this neuron across different syllable types ($n = 351$ syllables). Spike counts were calculated in a window from 50 ms before syllable onset to 20 ms after syllable onset. Datapoints are jittered for ease of view. A one-way ANOVA was run to assess whether different syllable types had significantly different mean spike counts ($F = 8.5$; $p = 1.5 \times 10^{-6}$). Horizontal red lines indicate the mean spike count for each syllable type, and vertical red lines indicate the standard error of the mean. **e** Scatter plot showing one-way ANOVA $p$-values and $F$-statistics for all significantly onset-locked NIf_{HVC} neurons recorded while birds listened to their tutor song. Horizontal red line shows Bonferroni-corrected significance threshold for $p = 0.05$. Vertical red line indicates $F$-statistic of 1.

electrode in HVC. Based on previous studies[88], neurons were identified as putative HVC-projectors if they had <100 μs latency jitter from HVC stimulation. Neurons were further identified as collision-tested HVC-projectors if they passed the collision test (triggering HVC stimulation on spontaneous spikes blocks stimulation-evoked spikes, see Supplementary Fig. 1).

**Analysis of song data**. Songs were recorded with custom MATLAB software (A. Andalman), which was configured to trigger recordings of all quiet vocalizations of juvenile birds. Songs were segmented into syllables using custom MATLAB software[40,69,74].

Syllable classification was performed using custom software[89] (based on ref. [90]), to visualize acoustic trajectories in low-dimensional space using t-stochastic

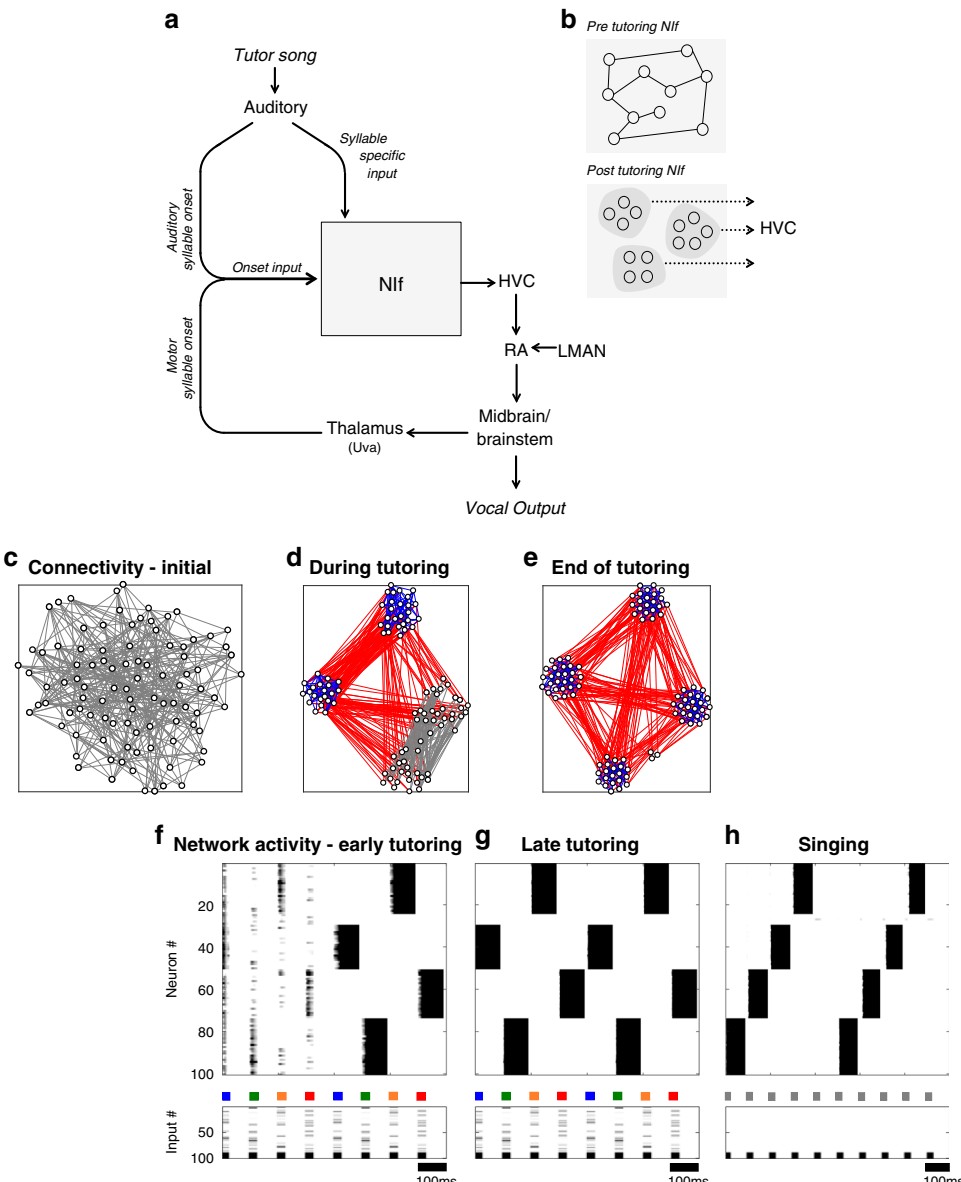

**Fig. 7 A neural network model for imprinting and reading out syllable-specific ensembles in Nlf. a** Schematic of model showing Nlf inputs and outputs. During tutoring, inputs include syllable-onset and syllable-specific signals that drive the formation of syllable-specific ensembles. During babbling, syllable-onset input, from an efference copy of vocal/respiratory commands, reactivates stored syllable ensembles. Reactivated Nlf ensembles then seed the development of syllable-specific motor sequences in HVC (see Fig. 8). **b** Illustration of ensemble formation in Nlf. Top: initial state, in which recurrent weights are random. Bottom: final state (after tutoring), in which recurrent weights produce self-reinforcing, but mutually inhibitory, ensembles. **c–e** Network connectivity at three stages of learning, displayed as a t-SNE embedding. Each point represents a neuron, and each line represents a connection between neurons. Weak connections are gray; excitatory connections are blue; inhibitory connections are red. For clarity, only the two strongest excitatory and inhibitory connections into each neuron are displayed. **c** Initial random connectivity, **d** Connectivity after early tutoring with four distinct syllables. **e** Connectivity at the end of tutoring. **f–g** Plots of network activity during tutoring. The Nlf network activity is shown at top, sorted so that ensembles appear as solid blocks; inputs are shown below, with colored bars indicating different syllables. Early in tutoring **f**, Nlf activity is relatively disorganized, but ensembles form quickly. Later in tutoring **g**, network activity has organized into four distinct ensembles selective for each tutor syllable. During singing **h**, Nlf ensembles are reactivated sequentially (gray bars indicate no syllable-specific inputs).

neighbor embedding (t-SNE) methods[91]. Syllable classification was confirmed by observing spectrograms of syllable renditions[40]. Syllable labeling was done without reference to neural activity.

**Analysis of neural data**. Many of our results involve assessing whether and how neural activity is aligned to syllable onsets. Spikes were sorted offline using custom MATLAB software (D. Aronov). In order to analyze the alignment of neural activity to syllable onsets, we calculated onset-aligned rate histograms (1 ms bins, smoothed over 20 bins). The latency of each unit was calculated as the time of the

onset-aligned PSTH maximum within a region from 100 ms before syllable onset to 100 ms after syllable onset[40].

In order to assess whether a neuron exhibited a significant modulation in response to syllables, we compared its PSTH to a temporally shifted control. Control rasters are generated by circularly shifting the timing of spikes by a different random (uniform from −100 to 100 ms) amount for each row (syllable) of the raster. In order to estimate $p$-values, the true PSTH is compared to PSTHs calculated from random control rasters. $p$-values are Bonferroni corrected for the number of neurons being tested. In the singing data, the PSTH peak was compared to peaks from random control PSTHs. In the tutoring data, where latencies were

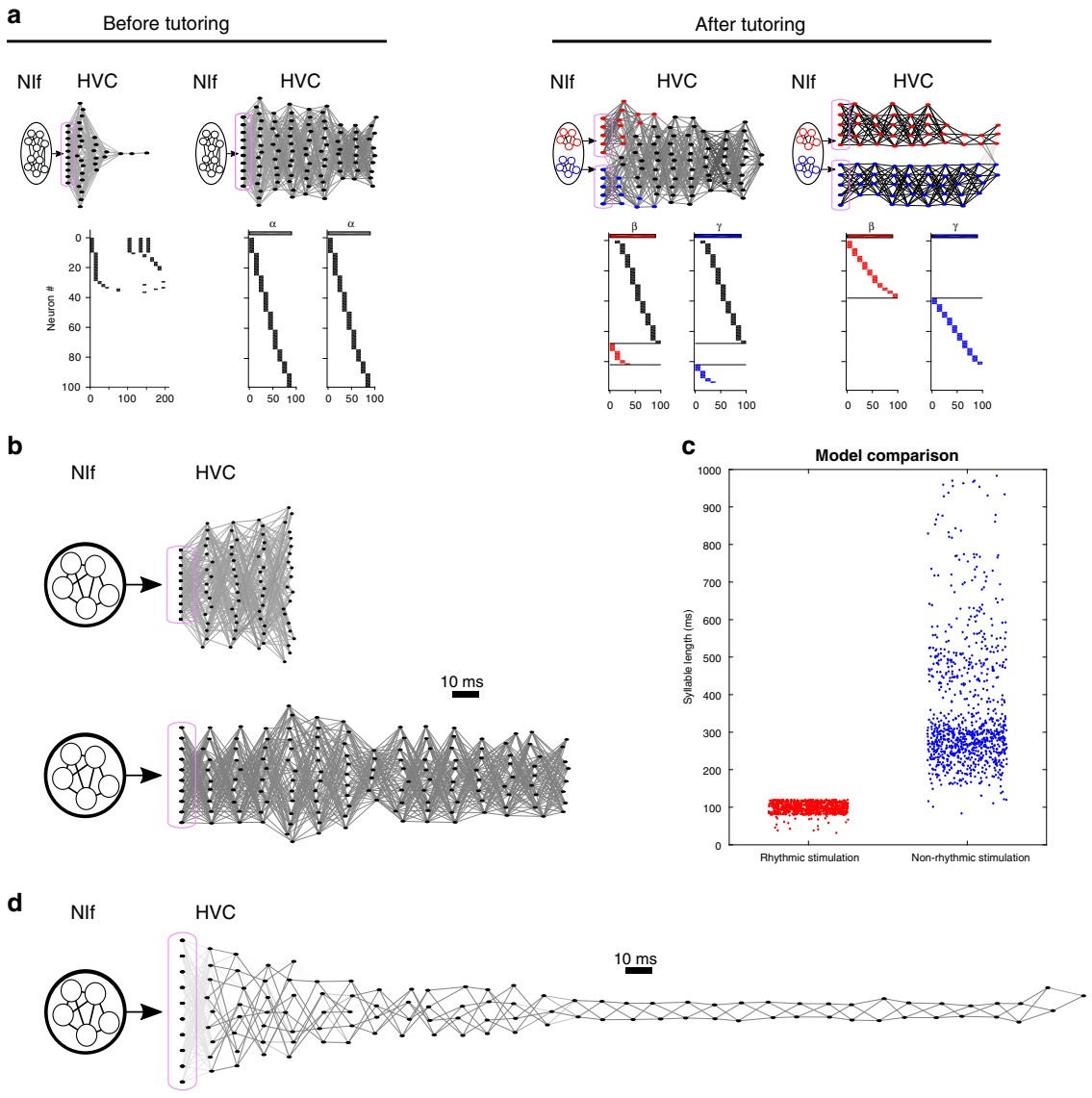

**Fig. 8 Combined NIf/HVC model. a** HVC model driven by NIf inputs into seed neurons in HVC, at four stages throughout learning (magenta boxes indicate seed neurons). Network diagrams of participating HVC neurons are shown at top (darker lines indicate stronger connections; and HVC neurons are sorted horizontally according to their latency relative to seed neurons). HVC network activity plots are shown below for example syllables. Left: before tutoring, NIf activity drives the formation of a single protosyllable chain in HVC. Right: after tutoring, clustered activity in NIf drives chain-splitting in HVC. **b** Rhythmic NIf input to seed neurons in HVC was driven at different periodicities: (top) 50 ms, and (bottom) 150 ms. Network diagrams show the HVC network after learning. **c** Length of syllables (duration of HVC activity) in the combined NIf/HVC model after the two stimulation protocols in Zhao et al. [62]: (red) Rhythmic NIf stimulation with period of 100 ms; (blue) non-rhythmic NIf stimulation. **d** An example HVC network diagram for a long chain (330 ms) produced by non-rhythmic NIf stimulation.

clustered in a narrow window 25 ms following syllable onset, the PSTH in this window was compared to control PSTHs in this window.

Population analyses combined syllable-aligned PSTHs across neurons. First, the PSTH of each neuron was normalized (mean subtracted; divided by standard deviation). Then, an average PSTH was calculated across the population. In order to assess the significance of the peak in the population response, a control set of PSTHs was generated by shifting the PSTH of each individual neuron by a different random amount (uniform between −100 and 100 ms) in time. Plots show shaded regions at the $p = 0.01$ level. Random controls are also used to assess the $p$-value of the peak population response. In several cases, the peak exceeded all $5 \times 10^6$ random controls; here $p$-values are stated as $p < 2 \times 10^{-6}$.

In order to analyze whether neurons were selective for particular syllable types, we performed ANOVA analyses comparing spike counts of the neuron across different syllable types. Spike counts were calculated in a window from 50 ms before syllable onset to 20 ms after syllable onset.

**Computational model**. MATLAB code used to simulate the model is available in Supplementary Code 1 in the supplementary information. Neurons were

modeled with activity as a threshold-linear function of membrane potential, with membrane potential capped at 0.5 to prevent runaway. Before each syllable presentation, membrane potential was reset to 0. Activity was simulated in continuous time, using MATLAB's ode45 (which uses a Runge–Kutta method) with a $dt$ of 1 ms.

A network of 100 of these neurons is recurrently connected in an all-to-all manner, with $\mathbf{W}_{ij}$ representing the synaptic strength from neuron $j$ to neuron $i$. Self-excitation is prevented by setting $\mathbf{W}_{ii} = 0$ for all $i$ at all times. In addition to recurrent input, each neuron receives a high dimensional feed-forward input. Input patterns are passed through an input weight matrix, with fixed synaptic weights drawn from an asymmetric distribution spanning positive and negative values. The distribution was constructed as a de-meaned log-normal distribution (standard deviation 0.25) chosen so there would be many weak inhibitory synapses and a few strong excitatory synapses. For example model weight matrices, see Supplementary Fig. 5.

There are two types of input to the network: first, a syllable-specific auditory input that is only active during the tutoring context, and second, a syllable onset signal that is active during both tutoring and singing. The first type of inputs were

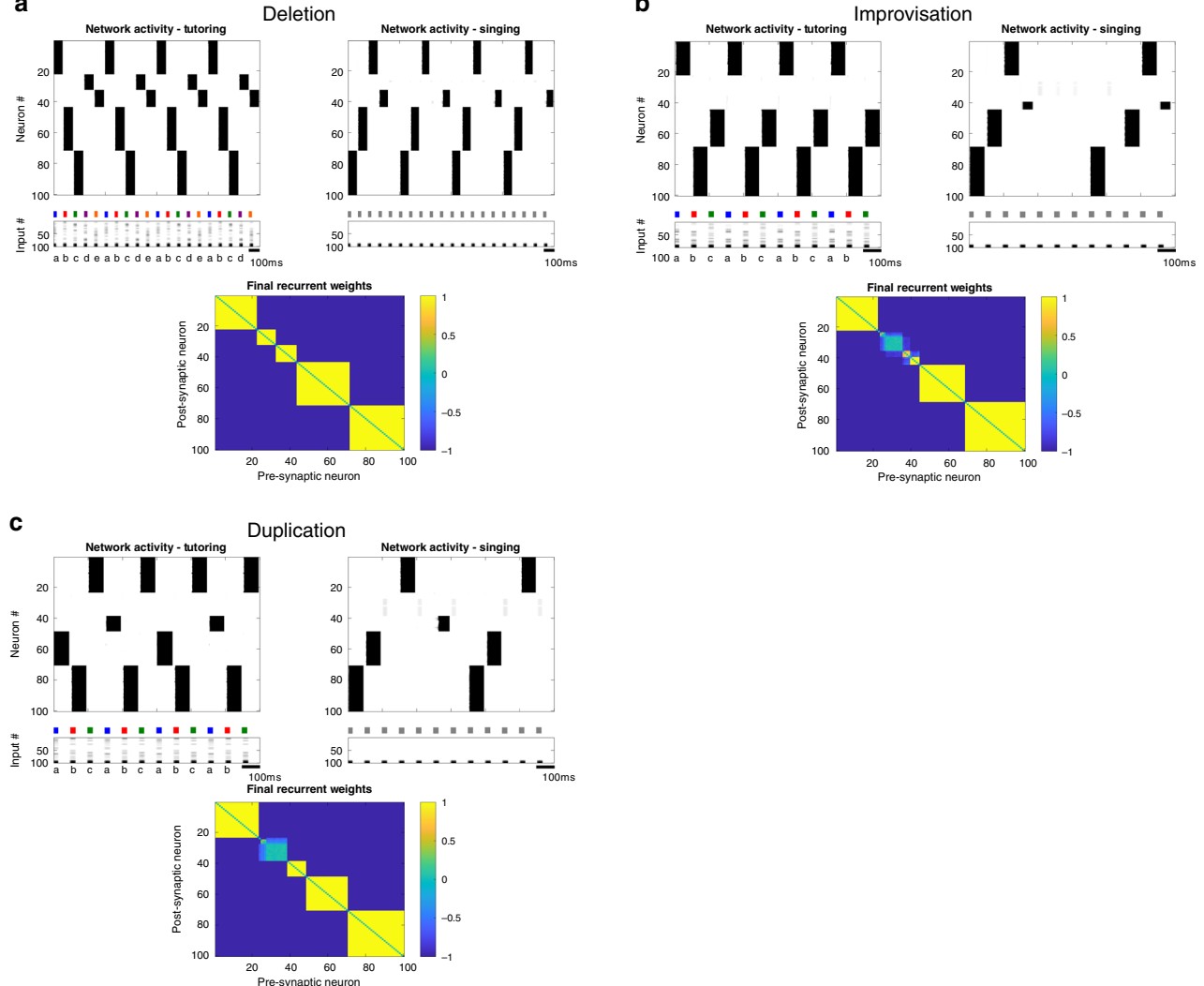

**Fig. 9 NIf model recapitulates interesting failure modes of vocal learning.** Under some circumstances, the model demonstrated failures of imitation, including deletion of a tutor syllable, improvised syllables, and duplicated syllables. Network activity during tutoring and singing, and final weight matrix, are shown for several interesting cases: **a** A case in which the network failed to reactivate, during the singing phase, an ensemble formed during tutoring (deletion). In this example, the network was tutored with five syllables. The network produced five ensembles during tutoring, but only four are successfully activated during singing. **b** A case in which the network spontaneously generates an un-tutored syllable during the singing phase (improvisation). Here, the network was tutored with three syllables. All three corresponding ensembles are activated during singing, as well as a fourth ensemble corresponding to leftover NIf neurons. **c** A case in which the network creates two distinct ensembles corresponding to the same tutor syllable (duplication). Here, the network was tutored with three syllables ('a', 'b', and 'c'). The network formed two different ensembles for tutor syllable 'a' (neurons 49–70, and neurons 39–48), activated on alternate presentations of the syllable 'a'. These two distinct ensembles are both replayed during singing, and in our model, would drive the formation of two different syllable sequences in HVC.

random (uniformly distributed between 0 and 1) sparse patterns in 100-dimensional space. Sparsity was achieved by forcing some proportion (80% unless otherwise specified) of the input elements to be 0. The second type of inputs were the same on every syllable type, and meant to represent a non-specific syllable onset signal to NIf. Each pattern is presented for 30 ms (30 timesteps), followed by 70 ms of silence. These patterns are presented sequentially for 20 cycles during a tutoring phase, during which the recurrent weights are updated. Then, during the singing phase, the full input patterns are replaced with the onset signal alone, repeated for 20 cycles. No learning occurs during the singing phase.

During the tutoring stage, these recurrent weights change according to a Hopfield-like learning rule, where $\mathbf{W}_{ij}$ is incremented by some $\Delta$ if neurons $i$ and $j$ are active, and $\mathbf{W}_{ij}$ is decremented by the same $\Delta$ if only one of neurons $i$ and $j$ is active. If both neurons are inactive, the synapse strength does not change. Each weight $\mathbf{W}_{ij}$ is capped to be between $+1$ and $-1$. For the anti-Hebbian learning, $\Delta = 0.05$, and for the Hopfield-like stage, $\Delta = 0.01$. Synapses are updated every time step.

During the first presentation of each syllable during tutoring, the recurrent weights undergo a phase of anti-Hebbian learning. That is, synapses between two co-active neurons are made more negative by an amount relative to the activity in each neuron. Specifically, $\Delta\mathbf{W} = -\eta_{\mathrm{AH}}(\mathbf{Y}_+\mathbf{Y}'_+)$, where $\eta_{\mathrm{AH}}$ is the anti-Hebbian learning rate (set to 0.05 unless stated otherwise), $\mathbf{Y}_+$ is the positive part of the membrane potential and $\mathbf{Y}'_+$ is the transpose of the positive part of the membrane potential. This is simply an outer product of the positive part of the membrane potential with itself, weighted by a learning rate. After each syllable is presented once, this anti-Hebbian learning ceases and is replaced by the Hopfield-like learning rule described above.

The network dynamics proceed according to the equation below:

$$\tau\dot{\mathbf{Y}}(t) = -\mathbf{Y}(t) + \mathbf{W}_+\mathbf{A}(t) + \mathbf{W}_-\mathbf{Y}_+(t) + \mathbf{W}_{\mathrm{B}}\mathbf{B}(t) - \boldsymbol{\alpha} - \boldsymbol{\Sigma} \qquad (1)$$

Where $\mathbf{Y}(t)$ is a vector containing the membrane potential for each neuron, $\mathbf{Y}_+(t)$ is the positive part of $\mathbf{Y}(t)$ (i.e., it is $\mathbf{Y}(t)$ where $\mathbf{Y}(t)$ is positive and 0 otherwise), $\tau$ is the membrane time constant (equal to 10 ms in all simulations), $\mathbf{W}_+$ and $\mathbf{W}_-$ are the positive and negative parts of the recurrent weight matrix, $\mathbf{B}(t)$ is a vector containing the inputs to the network, and $\mathbf{W}_{\mathrm{B}}$ is the input weight matrix. $\mathbf{A}(t)$ is the neural activity, which is a capped, threshold linear function of $\mathbf{Y}(t)$. $\boldsymbol{\alpha}$ is the

intracellular adaptation, which evolves (with time constant $\tau_i = 125$ ms and steady-state coefficient $\epsilon = 10$) according to

$$\tau_i \dot{\boldsymbol{\alpha}}(t) = \epsilon \mathbf{A}(t) - \boldsymbol{\alpha} \qquad (2)$$

Finally, $\boldsymbol{\Sigma}$ is a vector containing the normalization constant for each neuron. Each neuron is scaled by the average amount of feedforward inputs that it receives. $\boldsymbol{\Sigma}$ is calculated from the input weight matrix as follows: $\boldsymbol{\Sigma} = 0.75(\frac{1}{K}\sum_{i=1}^{K} \mathbf{W}_B \mathbf{B}_i)$, where $\mathbf{B}_i$ is the input pattern for syllable $i$, and $K$ is the number of syllables. The main function of this term is to generate clusters of relatively uniform sizes, and it also contributes to decorrelating different input patterns.

Most simulations were done with 100 neurons, but we wanted to check whether performance was similar for different network sizes. We found that the number of necessary training episodes is consistent across different network sizes (100, 500, and 1000 neurons, the order of magnitude of the total number of neurons in NIf, estimated from ref. [65]). The network learns quickly, from few training examples, consistent with birds' ability to learn from very limited exposure to a tutor song.

**t-SNE embedding plots**. Plots for Fig. 7 and the supplemental movies were created by performing t-SNE on the NIf model's recurrent weight matrix. To maximize similarity between embeddings across time, each run of t-SNE was given as initial conditions the embedding of the weight matrix during singing. MATLAB's tsne function was used. The only non-default parameter was exaggeration, which was set to 2.5.

Movie of network running was generated using the dynamic t-SNE algorithm from ref. [92], in order to preserve consistency of t-SNE embeddings across different frames.

**HVC model**. The HVC model was identical to that used in ref. [40]. Methods can be found there. In the section where rhythmic vs. non-rhythmic stimulation was compared, the model was driven with two different stimulus types. The rhythmic stimulation involved driving the model with trials composed of four pulses separated by 10 timesteps (100 ms total). The non-rhythmic stimulation involved driving the model with a single pulse. Inter-trial intervals were distributed according to a Poisson distribution with $\lambda = 50$ timesteps. The minimum inter-trial interval was 27 timesteps, and there were 7200 trials total. Trial structure was identical in rhythmic and non-rhythmic cases.

**Syllable length analysis**. At the end of these 7200 trials (see "HVC model" section above), the model was stimulated with a single pulse and allowed to run until activity ceased. The length of time for which the network was active following this stimulation was read-out as the syllable length. This single-pulse stimulation was repeated a total of 10 times for each run of the model, so that each run of the model produced 10 syllable length readings. This was done because the random noise in the network resulted in some variability in syllable length for each model run.

**Combined NIf/HVC model**. The combined model was achieved by driving the HVC model (described above and in ref. [40]) with inputs defined by the outputs of the NIf model (described above). To convert NIf model outputs into HVC model inputs, the following steps were taken. First, the NIf network was configured to learn two patterns (this is achieved by simply changing a parameter corresponding to the number of patterns to learn). This resulted in "singing phase" activity of the NIf network corresponding to two patterns. Then, since one timestep in the HVC model corresponds to 10 timesteps in the NIf model, this NIf output was downsampled across time by a factor of 10. Additionally, the NIf network (typically 100 neurons) is meant to drive a population of seed neurons (typically 10) in the HVC model. So, we also downsampled across neurons by a factor of 10. Finally, since the HVC model is driven by onsets, we filtered the NIf network output so that only activity onsets, rather than the entire activity, was non-zero.

In cases where the HVC model was stimulated with differing rhythms (i.e., every 5 or 15 timesteps), as in Fig. 8b, artificial stimulation of NIf-to-HVC neurons[62] was simulated by directly driving the HVC model with the frequencies described.

**Reporting summary**. Further information on research design is available in the Nature Research Reporting Summary linked to this article.

## Data availability
Data (raw electrophysiology and audio data, metadata, and annotations), as well as MATLAB code to generate data figures, is publicly available on the CRCNS data sharing platform (https://doi.org/10.6080/K0GQ6W0K)[93].

## Code availability
Analysis code and code to generate the data figures is posted alongside the raw data on the CRCNS data sharing platform (https://doi.org/10.6080/K0GQ6W0K)[93]. MATLAB code to run the model and generate figures related to the model is provided as supplementary material.

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

## Acknowledgements
This work was supported by a grant from the Simons Collaboration for the Global Brain, the National Institutes of Health (NIH) [grant number R01 DC009183] and the G. Harold & Leila Y. Mathers Charitable Foundation. E.L.M. received support through the NDSEG Fellowship program. Thanks to Andrew Bahle, Galen Lynch, Nader Nikbakht, Hannah Wirtshafter, and Leenoy Meshulam for comments on the manuscript.

## Author contributions
Conceptualization, E.L.M., M.T.L.H., and M.S.F.; Methodology, E.L.M., M.T.L.H., and M.S.F.; Software, E.L.M., and M.T.L.H.; Formal analysis, E.L.M., M.T.L.H., and M.S.F.; Investigation, E.L.M.; Writing—original draft, E.L.M., M.T.L.H., and M.S.F.; Writing—review & editing, E.L.M., M.T.L.H., and M.S.F.; Supervision, M.S.F.; Funding acquisition, M.S.F.

## Competing interests
The authors declare no competing interests.
