## [Peer Review File · Nature Communications]

Reviewers' Comments:

Reviewer #1:

Remarks to the Author:

This paper studies the role of NIf in song learning. It provides clear evidence that both during singing and tutoring NIf neurons burst selectively at syllable onsets. The authors go on to provide a mechanistic model of how such a network can form and show that the model's failure modes resemble that of vocal learning in songbirds. Previous studies from the authors on chain formation in HVC had predicted such syllable-specific neurons. Overall, this work provides a significant advance on the mechanistic understanding of motor learning in songbirds.

My main concern is about the model. Unfortunately authors did not provide sufficient information for a clear assessment. I list below some of the missing information. From what is given, I am concerned about the following points:

- 1) It isn't clear how the onset signal would lead to the same sequence in NIf as the tutor signal. Or should it not? If so, why?
- 2) Could the authors show an example of a simulation without the anti-Hebbian phase? What is the role of Σ in relation to anti-Hebbian learning? Which one is more important for decorrelating inputs? Why does one need both?
- 3) Simulations were done with 100 neurons. How does the number of necessary training episodes scale with network size, and how does this scaling compare to realistic network sizes and tutoring.

Missing information:

- 1) Bottom of page 18: What do you mean by a log-normal distribution with mean 0? A log-normal distribution is nonnegative. What is the variance of the log-normal distribution?
- 2) Please provide full detail about how the input patterns were created, e.g. what was the range of the uniform distribution, how was sparsity achieved.
- 3) What is the onset signal? How does it compare to the tutoring signal?
- 4) What value of Δ is used for synaptic plasticity? How often are the synapses updated? Every time step?
- 5) Can you provide the exact procedure and equations for the anti-Hebbian learning?
- 6) What is Y_{+} in the network dynamics equation? What is B? How is Σ calculated?
- 7) How is the capping done? On page 18 authors say they cap the membrane potential. On page 19 they say A is a capped function of Y. These are not the same thing.

Reviewer #2:

Remarks to the Author:

HVC neurons have been proved to show space pre-motor activity for specific syllables in a song and have been suggested to have local connection to form sequenced activity for producing stereo-typed song patterns. In this manuscript the authors tried to proof their hypothesis that the neuronal activity from the auditory system instruct the HVC sequence formation activity for song singing. To support this idea, the authors conducted in vivo

single unit electrophysiology recording from freely behaving zebra finch juveniles during they are singing or tutoring.

I found the experiments were well-designed and performed and analyzed in good quality, and it leveled Nif neurons showed pre-motor activity for each syllable, sometime to specific syllables). However, I think the experiments were just an observation of Nif neuronal firing correlated with zebra finch juvenile singing and tutor song hearing. It did not proof their hypothesis that Nif activity instruct HVC pre-motor singing activities. To answer their question raised this paper I think more direct experiment, such as testing if modifying or inactivating Nif activities would disrupt forming of HVC chunked pre-motor activity, would be needed.

Reviewer #3:

Remarks to the Author:

This study by Mackevius et al. is an exciting follow up study to the authors' 2015 paper characterizing the emergence of neural sequences underlying a complex, learned motor skill. As before, the authors use a pre-eminent model for investigating the neural mechanisms underlying vocal learning – the zebra finch – which offers the advantage of a well-defined neural circuit dedicated to song learning and production. Songbirds learn their vocalizations in two stages: first by memorizing the song of an adult 'tutor' and then by practicing their vocalizations. They use auditory feedback to evaluate their vocalizations and gradually modify their vocal output until they produce a good copy of the memorized tutor song. The prior study showed that a complex motor program gradually emerges from proto-sequences that split and differentiate.

Here, the authors ask how the song-generating motor circuit may be influenced during learning by the auditory system, focusing on the cortical nucleus NIf, which is at the interface between the motor and auditory systems. They record the activity of single neurons in NIf during singing and when the bird is listening to his tutor. Consistent with previous studies (Vyssotski et al.,2016), they find that NIf cells increase their firing during singing, mainly prior to syllable onsets, and can exhibit different firing patterns during different syllables. During tutoring, NIf neurons are activated at syllable onsets. Taking these two findings together, they propose a model in which tutor song exposure drives the formation of syllable-specific ensembles in NIf, which are re-activated during singing, providing inputs to train downstream premotor sequences. Thus, the model illuminates how learning in the auditory domain can guide learning in the motor domain, even capturing some of the common ways that songbirds fail to produce an accurate copy of the tutor song.

Comments below are aimed at strengthening the work through clarification of key concerns:

Major Comments:

1) My main concern with this study is with the analysis of auditory responses in NIf during tutoring. The authors recorded the activity of NIf neurons when birds were listening to the tutor singing, and found that some NIf neurons fired preferentially after the onset of tutor

syllables. Moreover, individual neurons could respond differentially to different syllables, consistent with the idea that exposure to the tutor song drives formation syllable specific ensembles in NIf.

However, the finding that NIf neurons burst preferentially at tutor song syllable onsets is strongly overstated. First, while the authors assert that 9/13 NIfHVC neurons exhibited tutor-locked increases in firing rate (line 209), this is not apparent in the group data. Fig. 5E shows that 5 neurons exhibited a clear increase in activity $\sim 0-40$ ms after syllable onset (cells with red bursts after syllable onset in Fig. 5E). Just as many neurons, however, showed either no change in firing rate (white) or a decrease in mean firing rate (blue; $n=4$ cells) after the onset of the tutor syllable, but the authors do not mention such inhibitory responses. The proposed computational model emphasizes the contribution of excitatory signals aligned to syllable onsets for facilitating syllable-specific ensembles, but it would be useful to add some discussion of whether and how inhibitory responses could contribute to chunking the song' in the listening context.

Second, it is not clear whether NIf neurons also encode the offset of tutor song syllables (i.e., increase firing at the end of tutor syllable). A subset of NIf neurons fired bursts before the onset of tutor syllables (Figs 5G , 5E), and the authors hypothesized that these neurons may be responding to the offset of the previous syllable. To examine this issue directly, analysis of syllable offset-aligned rasters and rate histograms would help to determine whether increases in firing rate occur in a reasonable "auditory" window after tutor syllable offsets. While the authors did try to decouple syllable onsets and syllable offsets using an artificial stimulus with gap durations and syllable durations drawn from a random distribution (Fig. 13), it remains unclear whether NIf neurons exhibit offset responses to highly stereotyped, predictable tutor song.

In addition, it would be interesting to examine whether an individual neuron exhibits both onset- and offset-aligned bursts (within a reasonable "auditory window"), whether neurons exhibit differential offset responses depending on syllable type, and/or whether different subsets of neurons code for syllable onsets and offsets.

The notion that both onset-aligned responses and offset-aligned responses in NIf neurons might facilitate formation of syllable specific ensembles seems plausible given recent findings from the Roberts lab showing that optogenetic stimulation of NIf inputs to HVC can affect syllable duration. Zhao et al (2019) showed that stimulation with a short sequence of 50ms pulses resulted in songs with short syllables, while stimulation with a 300ms light pulse resulted in songs with significantly longer syllables. While it is not clear how the light pulses affect the firing patterns in HVC, the finding that a long light pulse drives production of longer syllables suggest that both the onset and offset of NIf inputs to HVC may be important for syllable specific ensemble formation.

Other Comments:

2) The proposed model involves an arbitrary change in the learning rule - a brief period of

“anti-Hebbian learning” during which synapses of co-active neurons are weakened followed by a “Hopfield-like Hebbian learning rule”. It would be helpful if the authors could provide justification for this choice and/or cite references for known examples of similar changes in the learning rules.

3) The model includes two types of inputs: i) a syllable-specific auditory input during tutoring; and ii) a syllable onset signal during tutoring and singing. According to the model, the syllable onset signal during singing corresponds to an efference copy of the motor commands originating in brain areas responsible for the initiation of babbling. The authors suggest that the efference copy originates in a cortical nucleus LMAN and reaches NIf via a feedback pathway from the midbrain via the thalamic nucleus Uva. However, from Figure 1, it is clear that there may be multiples sources of the efference copy from LMAN to the premotor nucleus HVC: Uva sends a direct projection to HVC, an indirect projection via the auditory cortical region CM, and an indirect projection via NIf. It would be helpful to include some discussion of whether the other inputs (Uva to HVC and Uva to CM to HVC) could also contribute to facilitating chains of sequentially active neurons in HVC. For example, how does the singing-related activity or auditory responses of differ between HVC projectors in Uva and HVC projectors in NIf? Could the multiple efference copies (and variable timing of their arrival at HVC) help explain the multiple failure modes of vocal learning?

4) At several places in the manuscript, the sample size is not clear or does not match with the text:

Figure 6E shows data for all NIfHVC neurons during tutoring. In the figure, there seem to be 19 data points, but the text states that 13 putative HVC-projectors were recorded during tutoring (lines 189-190). Lines 190 and 201-202 state that 9 HVC projectors are tutor-locked, but there are 8 data points below the red line indicating a significant change in the mean firing rate during listening. Also, the color coding in Figure 6E is not clear. In Fig. 4F, green indicates NIfHVC neurons, but Fig. 6E only shows data for HVC projectors. Please verify and clarify the sample sizes.

Figure 11/Supplementary Figure – please state the number of cells and birds. Figure 11 shows a subset of the total data set (4 birds, not 7 with antidromically identified neurons or 9 total birds) and does not mention how many cells are HVC projectors vs. putative interneurons. This latter point is important because the authors want to assure readers that the narrow distribution of NIf latencies is not an artifact of sampling given that projections from NIf to HVC are not uniform through the nucleus, and different regions in HVC may have distinct roles.

5) Bursts are never defined. Did the authors use a firing rate threshold or the ISI distribution to define bursts?

6) Fig. 9/ Supp. Fig – related to – Figure 1?

7) Figs. 2E & 5E – missing the “Different colored squares” that indicate different birds.

8) Figs. 3c & 3d – x-axis is missing time before 0ms (syllable onset).

Response to referee letter on 'Chunking song into syllables: A cortical circuit for translating tutor song into simple vocal-motor units', co-authored by EL Mackevicius, MTL Happ, and MS Fee

Reviewer #1 (Remarks to the Author):

This paper studies the role of Nlf in song learning. It provides clear evidence that both during singing and tutoring Nlf neurons burst selectively at syllable onsets. The authors go on to provide a mechanistic model of how such a network can form and show that the model's failure modes resemble that of vocal learning in songbirds. Previous studies from the authors on chain formation in HVC had predicted such syllable-specific neurons. Overall, this work provides a significant advance on the mechanistic understanding of motor learning in songbirds.

My main concern is about the model. Unfortunately authors did not provide sufficient information for a clear assessment. I list below some of the missing information. From what is given, I am concerned about the following points:

We thank the reviewer for highlighting the need for more clarity about several aspects of the model. We have substantially clarified our description of the model, and added several exciting new modeling results, based on reviewer suggestions.

1) It isn't clear how the onset signal would lead to the same sequence in Nlf as the tutor signal. Or should it not? If so, why?

We apologize for the lack of clarity -- the onset signal does not lead to the same syllable order during the singing stage as presented during the tutor stage. We did not set out to model how the syllable order is represented or produced by the songbird brain. The syllable order was only occasionally identical during singing and tutoring (16/100 random initializations of the model). However, we did find that within the singing phase, the syllable order was typically consistent across different renditions (91/100 random initializations of the model), since the neuronal adaptation makes recently active neurons less likely to get activated again, leading the network to cycle through syllables, instead of repeating the first syllable. We have clarified the text on this point.

2) Could the authors show an example of a simulation without the anti-Hebbian phase? What is the role of Σ in relation to anti-Hebbian learning? Which one is more important for decorrelating inputs? Why does one need both?

We thank the reviewer for raising this important point for clarification. To further investigate the role of anti-Hebbian learning and synaptic normalization in the model, we simulated the model without each of them, and found that the two factors appear to have somewhat different roles. When both the anti-Hebbian phase and the normalization are included, the model successfully produces the correct number of syllables 98% of the time, compared to <10% without either factor. With only anti-Hebbian learning, but no normalization, model performance was only very slightly degraded (96% success). However, the sizes of different clusters were much less uniform. Thus, the important role of synaptic normalization in the model was to keep the sizes of clusters relatively uniform. On its own, synaptic normalization also had moderate success in decorrelating input patterns (80% success), but the main motivation for including it in the model was to keep cluster sizes uniform. Anti-Hebbian learning played a more major role in decorrelating input patterns, as has been observed in other models (Pehlevan et al. 2018; Foldiak 1990). We have added an explanation of this to the text.

3) Simulations were done with 100 neurons. How does the number of necessary training episodes scale with network size, and how does this scaling compare to realistic network sizes and tutoring.

We found that the number of necessary training episodes is consistent across different network sizes (100, 500, and 1000 neurons, the latter being approximately the total number of neurons in Nlf, estimated from Elliot et al. 2017). The network learns quickly from few training examples, consistent with birds' ability to learn from very limited exposure to a tutor song. We have added a description of these results to the text.

Missing information:

1) Bottom of page 18: What do you mean by a log-normal distribution with mean 0? A log-normal distribution is nonnegative. What is the variance of the log-normal distribution?

We thank the reviewer for pointing out the sloppy terminology here related to the distribution of input weights. These fixed synaptic weights were drawn from an asymmetric distribution spanning positive and negative values. The distribution was constructed as a de-meanded log-normal distribution (standard deviation 0.25) chosen so there would be many weak inhibitory synapses and a few strong excitatory synapses. We have clarified in the text.

2) Please provide full detail about how the input patterns were created, e.g. what was the range of the uniform distribution, how was sparsity achieved.

This has been clarified in the methods. The input patterns were first drawn from a uniform distribution with values between 0 and 1, then sparsity was achieved by forcing 80% of the input elements to be 0.

3) What is the onset signal? How does it compare to the tutoring signal?

The tutoring signal is an input pattern that is specific to each syllable. The onset signal is identical for all syllables. Both the tutoring and onset signals are fed to the network during the “tutoring phase,” but only the onset signal is fed to the network during the “singing” phase. We have clarified the description in the methods.

4) What value of Δ is used for synaptic plasticity? How often are the synapses updated? Every time step?

Δ is 0.01 for Hopfield-like learning and 0.05 for anti-Hebbian learning. Synapses are updated at each time step. We have added this information to the methods.

5) Can you provide the exact procedure and equations for the anti-Hebbian learning?

The equation for the anti-Hebbian learning is: $\Delta W = -\eta_{AH} (Y_{+} Y_{+}')$, where η_{AH} is the anti-Hebbian learning rate (set to 0.05 unless stated otherwise), Y_{+} is the positive part of the membrane potential and Y_{+}' is the transpose of the positive part of the membrane potential. This is simply an outer product of the positive part of the membrane potential with itself, weighted by a learning rate.

We have added this information to the methods.

6) What is Y_{+} in the network dynamics equation? What is B? How is Σ calculated?

We thank the reviewer for pointing out these omissions from the methods, and we have now included them.

Y_{+} is the positive part of Y (i.e., it is Y where Y is positive and 0 otherwise).

B is a vector containing the inputs to the network.

Σ is a vector that scales each neuron by the average amount of feedforward inputs that it receives: $\Sigma = 0.75(1/K \sum_{i=1}^K W_{B_i})$, where B_i is the input pattern for syllable i , and K is the number of syllables.

7) How is the capping done? On page 18 authors say they cap the membrane potential. On page 19 they say A is a capped function of Y . These are not the same thing.

Only the activity (A) is capped, not the membrane potential. Capping is done by replacing the calculated value with $\max(\text{cap}, \text{calculated value})$. We apologize for the mistake, and have now clarified this in the methods.

Reviewer #2 (Remarks to the Author):

HVC neurons have been proved to show space pre-motor activity for specific syllables in a song and have been suggested to have local connection to form sequenced activity for producing stereo-typed song patterns. In this manuscript the authors tried to proof their hypothesis that the neuronal activity from the auditory system instruct the HVC sequence formation activity for song singing. To support this idea, the authors conducted in vivo single unit electrophysiology recording from freely behaving zebra finch juveniles during they are singing or tutoring.

I found the experiments were well-designed and performed and analyzed in good quality, and it leveled Nif neurons showed pre-motor activity for each syllable, sometime to specific syllables). However, I think the experiments were just an observation of Nif neuronal firing correlated with zebra finch juvenile singing and tutor song hearing. It did not proof their hypothesis that Nif activity instruct HVC pre-motor singing activities. To answer their question raised this paper I think more direct experiment, such as testing if modifying or inactivating Nif activities would disrupt forming of HVC chunked pre-motor activity, would be needed.

We appreciate the reviewer's focus on perturbation experiments. Indeed, several key perturbation experiments have recently been done in other labs, and are highly relevant to our findings and model. Lesions of Nif have minimal effect on adult zebra finch song (Otchy et al., 2015), but affect syllable ordering (Hosino et al., 2000) in Bengalese finches. Inactivation of Nif in young juvenile birds causes loss of emerging spectral and temporal song structure (Naie et al., 2011), and inactivating Nif while a juvenile bird is being tutored interferes with tutor imitation (Roberts et al., 2012}. It was recently

observed that optogenetic stimulation of Nlf can influence the duration of eventual song syllables (Zhao et al., 2019).

We would point out that, while optogenetics and inactivation experiments are important, neural recordings are extremely important. (Indeed, it has been noted that by themselves, perturbation experiments have some limitations in elucidating how complex neural systems function; Otchy 2015 and Jazayeri 2017). This manuscript presents the first recordings of Nlf neurons during tutoring, and the most complete recordings of Nlf during singing. Our manuscript also contains a well-developed theoretical interpretation that captures the neurophysiology as well as causal experiments.

The reviewer's comments motivated us to take a closer look at the predictive power of our theoretical framework, in light of newly published optogenetics experiments in Nlf. First, we connected the Nlf model to a model of HVC, and found that activity in Nlf was able to train HVC to assemble new sequences for each 'tutored' syllable. The optogenetics results of Zhao et al. provide a rare opportunity to test our theoretical framework, since they were published after we submitted this manuscript, and did not inform our modeling work.

Zhao et al. found that patterned optogenetic stimulation of Nlf in young birds was able to affect the durations of learned song syllables. For example, stimulating Nlf at a fast 10Hz rhythm led birds to sing shorter syllables (Zhao et al., 2019), while stimulation of Nlf with longer pulses (300ms) at irregular intervals led to the formation of pathologically long syllables with a wide range of durations (from 150ms to almost one second). We found that our combined Nlf/HVC model precisely reproduces these findings: rhythmic Nlf activation led to the formation of precisely timed syllables with a duration equal to the period of the stimulation, while irregular stimulation at long intervals led to highly variable durations of HVC network activity, corresponding to long syllables ranging from 150ms up to a second in duration.

The combined Nlf/HVC model significantly expanded the predictive power of our theoretical framework, and we have added a figure describing these new results. We thank the reviewer for the suggestion to consider specific perturbation experiments in the context of this framework.

Reviewer #3 (Remarks to the Author):

This study by Mackevius et al. is an exciting follow up study to the authors' 2015 paper characterizing the emergence of neural sequences underlying a complex, learned motor

skill. As before, the authors use a pre-eminent model for investigating the neural mechanisms underlying vocal learning – the zebra finch – which offers the advantage of a well-defined neural circuit dedicated to song learning and production. Songbirds learn their vocalizations in two stages: first by memorizing the song of an adult ‘tutor’ and then by practicing their vocalizations. They use auditory feedback to evaluate their vocalizations and gradually modify their vocal output until they produce a good copy of the memorized tutor song. The prior study showed that a complex motor program gradually emerges from proto-sequences that split and differentiate.

Here, the authors ask how the song-generating motor circuit may be influenced during learning by the auditory system, focusing on the cortical nucleus NIf, which is at the interface between the motor and auditory systems. They record the activity of single neurons in NIf during singing and when the bird is listening to his tutor. Consistent with previous studies (Vyssotski et al., 2016), they find that NIf cells increase their firing during singing, mainly prior to syllable onsets, and can exhibit different firing patterns during different syllables. During tutoring, NIf neurons are activated at syllable onsets. Taking these two findings together, they propose a model in which tutor song exposure drives the formation of syllable-specific ensembles in NIf, which are re-activated during singing, providing inputs to train downstream premotor sequences. Thus, the model illuminates how learning in the auditory domain can guide learning in the motor domain, even capturing some of the common ways that songbirds fail to produce an accurate copy of the tutor song.

Comments below are aimed at strengthening the work through clarification of key concerns:

Major Comments:

1) My main concern with this study is with the analysis of auditory responses in NIf during tutoring. The authors recorded the activity of NIf neurons when birds were listening to the tutor singing, and found that some NIf neurons fired preferentially after the onset of tutor syllables. Moreover, individual neurons could respond differentially to different syllables, consistent with the idea that exposure to the tutor song drives formation syllable specific ensembles in NIf.

However, the finding that NIf neurons burst preferentially at tutor song syllable onsets is strongly overstated. First, while the authors assert that 9/13 NIfHVC neurons exhibited tutor-locked increases in firing rate (line 209), this is not apparent in the group data. Fig. 5E shows that 5 neurons exhibited a clear increase in activity ~0-40ms after syllable onset (cells with red bursts after syllable onset in Fig. 5E). Just as many neurons,

however, showed either no change in firing rate (white) or a decrease in mean firing rate (blue; $n=4$ cells) after the onset of the tutor syllable, but the authors do not mention such inhibitory responses. The proposed computational model emphasizes the contribution of excitatory signals aligned to syllable onsets for facilitating syllable-specific ensembles, but it would be useful to add some discussion of whether and how inhibitory responses could contribute to chunking the song' in the listening context.

The reviewer raised an important concern about the robustness of our claim about syllable onset locking in Nlf during tutoring. To address this concern, we have now carried out a new, more careful analysis of Nlf activity, including analysis of additional data from identified projecting neurons and non-identified neurons. The new analysis and data strongly support our original conclusions about syllable-onset locking.

Several of the reviewer's comments have led us to rethink our initial criteria for including or excluding recorded neurons based on the amount of data (i.e. recording duration). We previously used an inclusion criterion based on an overly-stringent statistical test of the reliability of the syllable-onset PSTHs. This led to the exclusion of 16 identified projection neurons and 38 non-identified neurons, all of which were recorded during at least 180 tutor song syllables, more than enough to detect a syllable-onset locked modulation.

We have decided to include in the dataset all Nlf neurons recorded during tutoring (29 identified HVC-projectors and 74 non-projectors). The non-identified neurons were all recorded within the borders of Nlf, as determined by the existence of antidromically activated 'hash' in the immediate region of the recordings. The main Figure 6 now shows data from all 103 Nlf neurons in the tutoring dataset.

Putting all of these neurons together now, there is strong evidence for onset responses in Nlf during tutoring at both the population level and at the individual neuron level. The population average syllable-aligned PSTH exhibited a significant peak in activity 14ms after syllable onset (Figure 5E , top; $p < 2e^{-6}$). The post-onset peak was observed separately in both the population of Nlf_{HVC} projection neurons (peak latency 2ms; $p < 1e^{-4}$) and in the population of non-identified neurons (peak latency 15ms; $p < 2e^{-6}$). A more detailed analysis of individual neurons revealed that 51 of all 103 neurons (including 8 of 29 projection neurons) exhibited significantly elevated firing rates in a window between 0 and 25ms after tutor syllable onset ($p < 0.05$, Bonferroni corrected).

The reviewer also raises the interesting question of whether any neurons exhibited a significant decrease in firing rate at tutor syllable onsets. A similar analysis revealed that, in fact, 6 of 103 neurons (including 1 HVC-projector) exhibited a statistically significant decrease in firing rate in the 0 to 25ms window after syllable onset (Bonferroni corrected for 103 comparisons). For these neurons, the average firing rate change in this window was -7Hz, compared to the +22Hz firing rate change for the 51 neurons exhibiting an increase. Thus, as a population, the predominant modulation in Nlf was a brief increase in firing rate after syllable onsets. We have added these new results about significant firing rate dips to the manuscript.

Second, it is not clear whether Nlf neurons also encode the offset of tutor song syllables (i.e., increase firing at the end of tutor syllable). A subset of Nlf neurons fired bursts before the onset of tutor syllables (Figs 5G , 5E), and the authors hypothesized that these neurons may be responding to the offset of the previous syllable. To examine this issue directly, analysis of syllable offset-aligned rasters and rate histograms would help to determine whether increases in firing rate occur in a reasonable “auditory” window after tutor syllable offsets. While the authors did try to decouple syllable onsets and syllable offsets using an artificial stimulus with gap durations and syllable durations drawn from a random distribution (Fig. 13), it remains unclear whether Nlf neurons exhibit offset responses to highly stereotyped, predictable tutor song.

In addition, it would be interesting to examine whether an individual neuron exhibits both onset- and offset-aligned bursts (within a reasonable "auditory window"), whether neurons exhibit differential offset responses depending on syllable type, and/or whether different subsets of neurons code for syllable onsets and offsets.

The reviewer raises the very interesting question of what role syllable offsets may play in driving Nlf activity. We thus directly analyzed Nlf responses to tutor syllable offsets. For most syllables in the tutor song, it is difficult to completely decouple syllable onset vs offset responses, because the offset of most syllables is followed by the onset of the next song syllable. Therefore to analyze syllable offsets, we aligned Nlf activity to the offsets of the final syllable in the tutor song bout. As a population, Nlf neurons exhibited no significant modulation (neither excitation nor inhibition) following last syllable offset. In an individual neuron analysis, only 2/103 neurons exhibited a significant increase in spiking activity in a 0-25ms window following last-syllable offset. We have included onset- and offset- aligned rasters for one of these neurons (the one with the most robust response) in supplemental figure 13. Interestingly, both of these neurons also exhibited a robust syllable-onset response. The description of offset activity has been included in

Supplementary Materials. Based on this negative result, we feel it would be most appropriate to remove the relatively unconvincing artificial subsong results previously shown in Supp Fig 13.

The notion that both onset-aligned responses and offset-aligned responses in Nlf neurons might facilitate formation of syllable specific ensembles seems plausible given recent findings from the Roberts lab showing that optogenetic stimulation of Nlf inputs to HVC can affect syllable duration. Zhao et al (2019) showed that stimulation with a short sequence of 50ms pulses resulted in songs with short syllables, while stimulation with a 300ms light pulse resulted in songs with significantly longer syllables. While it is not clear how the light pulses affect the firing patterns in HVC, the finding that a long light pulse drives production of longer syllables suggest that both the onset and offset of Nlf inputs to HVC may be important for syllable specific ensemble formation.

The reviewer raises the very interesting question of how syllable durations may be controlled by inputs to HVC from Nlf, pointing to recent optogenetic manipulations in Nlf by Zhao et al. More specifically, Zhao et al. found that patterned optogenetic stimulation of Nlf in young birds was able to affect the durations of learned song syllables. For example, stimulating Nlf at a fast 10Hz rhythm led birds to sing shorter syllables (Zhao et al., 2019), while stimulation of Nlf with longer pulses (300ms) at irregular intervals led to the formation of pathologically long syllables with a wide range of durations (from 150ms to almost one second).

One possibility, pointed out by the reviewer, is that Nlf may explicitly encode both syllable onsets and offsets. Or that HVC may be responsive to the offsets of Nlf inputs. However, our data are not consistent with the idea that Nlf significantly codes for syllable offsets, and are not consistent with the idea that Nlf encodes syllables with long periods of persistent activation.

We consider an alternative hypothesis that syllable durations are controlled by a combination of Nlf inputs and dynamics within the HVC network. In other words, Nlf may activate HVC, which then 'reverberates' for a period of time until it is either reset by the next Nlf input, or the reverberation in HVC dies out, as determined by its intrinsic recurrent excitation and inhibition. We took this opportunity to explicitly examine this latter hypothesis using a combined Nlf/HVC model.

We found that our combined Nlf/HVC model precisely reproduces these findings: rhythmic Nlf activation led to the formation of precisely timed syllables with a duration equal to the period of the stimulation, while irregular stimulation at long intervals led to

highly variable durations of HVC network activity, corresponding to long syllables ranging from 150ms up to a second in duration. In summary, we found that we were able to replicate the Zhao optogenetic results in our combined Nlf/HVC model, which combines syllable onset signals from Nlf with recurrent network dynamics in HVC.

Other Comments:

2) The proposed model involves an arbitrary change in the learning rule - a brief period of “anti-Hebbian learning” during which synapses of co-active neurons are weakened followed by a “Hopfield-like Hebbian learning rule”. It would be helpful if the authors could provide justification for this choice and/or cite references for known examples of similar changes in the learning rules.

Several recent lines of work have suggested learning rules that may group networks of neurons into clusters that reflect their inputs: Pehlevan et al. 2018 and Litwin-Kumar et al. 2015. Both lines of work agree that qualitatively similar network clusters can be achieved through a variety of different learning mechanisms. However, it appears that the mechanisms that work share two key principles: some form of Hebbian plasticity to group neurons together; and some form of anti-Hebbian competition to separate different clusters. Therefore, we chose to include both. In our model, we found that each form of learning was only necessary at one particular stage of learning.

We thank the reviewer for bringing up this point, and have incorporated the above references and discussion into our introduction of the model.

3) The model includes two types of inputs: i) a syllable-specific auditory input during tutoring; and ii) a syllable onset signal during tutoring and singing. According to the model, the syllable onset signal during singing corresponds to an efference copy of the motor commands originating in brain areas responsible for the initiation of babbling. The authors suggest that the efference copy originates in a cortical nucleus LMAN and reaches Nlf via a feedback pathway from the midbrain via the thalamic nucleus Uva. However, from Figure 1, it is clear that there may be multiples sources of the efference copy from LMAN to the premotor nucleus HVC: Uva sends a direct projection to HVC, an indirect projection via the auditory cortical region CM, and an indirect projection via Nlf. It would be helpful to include some discussion of whether the other inputs (Uva to HVC and Uva to CM to HVC) could also contribute to facilitating chains of sequentially active neurons in HVC. For example, how does the singing-related activity or auditory responses of differ between HVC projectors in Uva and HVC projectors in Nlf? Could

the multiple efference copies (and variable timing of their arrival at HVC) help explain the multiple failure modes of vocal learning?

We thank the reviewer for suggesting these discussion points. All of these seem like reasonable possibilities, and we have added them to the manuscript.

4) At several places in the manuscript, the sample size is not clear or does not match with the text:

We apologize for the sloppiness, and have corrected it.

Figure 6E shows data for all NlfHVC neurons during tutoring. In the figure, there seem to be 19 data points, but the text states that 13 putative HVC-projectors were recorded during tutoring (lines 189-190). Lines 190 and 201-202 state that 9 HVC projectors are tutor-locked, but there are 8 data points below the red line indicating a significant change in the mean firing rate during listening. Also, the color coding in Figure 6E is not clear. In Fig. 4F, green indicates NlfHVC neurons, but Fig. 6E only shows data for HVC projectors. Please verify and clarify the sample sizes.

The reviewer noticed an inconsistency between sample sizes reported in Figure 6E and the text. This led us to realize we were running a different shuffled control for some of our analyses. We have re-run all of our analyses and figures with the procedures described in the manuscript. We thank the reviewer for catching this, and we have now verified and clarified the sample sizes.

Regarding Figure 6E, we have corrected the sample sizes and clarified the legend.

Figure 11/Supplementary Figure – please state the number of cells and birds. Figure 11 shows a subset of the total data set (4 birds, not 7 with antidromically identified neurons or 9 total birds) and does not mention how many cells are HVC projectors vs. putative interneurons. This latter point is important because the authors want to assure readers that the narrow distribution of Nlf latencies is not an artifact of sampling given that projections from Nlf to HVC are not uniform through the nucleus, and different regions in HVC may have distinct roles.

As suggested, we have now marked which neurons were HVC projectors, and stated the number of cells and birds (33 neurons, including 11 projectors from 6 birds). We have clarified that the data included in this figure are the cases where: (1) it was possible to unambiguously estimate in the histology which track corresponded to which

recording electrode, and (2) the song-locked PSTH had a significant peak, so it was possible to estimate the neuronal latency.

5) Bursts are never defined. Did the authors use a firing rate threshold or the ISI distribution to define bursts?

We have clarified the text. In this paper, we don't use burst as a quantitative concept, but rather as a figure of speech to describe a brief increase in firing rate. Burst latency is defined as the time of the peak in the syllable-onset-aligned PSTH.

6) Fig. 9/ Supp. Fig – related to – Figure 1?

Yes. We have corrected the typo.

7) Figs. 2E & 5E – missing the “Different colored squares” that indicate different birds.

We removed the colored squares from the figure because we felt they added unnecessary clutter, but forgot to delete this text from the caption. This has been fixed.

8) Figs. 3c & 3d – x-axis is missing time before 0ms (syllable onset).

The axis was there, but the axis labels were typeset in a confusing way, which we have now fixed.

Reviewers' Comments:

Reviewer #1:

Remarks to the Author:

The authors thoroughly addressed all my concerns. I have no further comments.

Reviewer #2:

Remarks to the Author:

In response to my request, providing experimental data for showing Nif instructing HVC neurons, they provided new modeling, Nif-HVC combined modeling based on their Nif recording data and recent publication from the other lab. That did not provide a new conceptual idea, however their Nif recording is importance to test if the modeling concept is reasonable and their experiments and model well confirmed the suggested concept. However, here is my additional comments on the additional parts.

The authors stated "The latencies of individual NIfHVC neurons are clustered prior to syllable onsets at all stages of song learning, with 80%, 79% and 83% of bursts occurring prior to syllable onset in subsong, protosyllable and multisyllable stages respectively (Figure 3C). In contrast, the distribution of HVC burst latencies appears to change over development in a way consistent with chains growing from syllable onsets [40]. In adult birds, HVC bursts occur fairly uniformly throughout the song motif [38, 39]." (l164) However, it was not clear why it can be concluded in the following sentence "These results are consistent with a model in which HVC sequences grow from syllable onsets, triggered by inputs from NIf (Figure 3E)." More importantly, it was not stated well how these developmental changes observed in the NIf recording was implemented in their Nif-HVC combined modeling. Did authors want to suggest that connectivity between Nif-HVC alters over development (latency changes could be explained by this?),.

The authors further stated that "This model demonstrates explicitly how the formation of different ensembles in NIf can drive chain splitting in HVC such that different syllables in the song are represented by different chains in HVC" (L357). However, they did not explain how the experimental observation over the development match with the modeling.

The authors stated "Of the neurons we were able to record 224 during both singing and tutoring, some were exclusively singing-locked (12/24 225 neurons, including 7/10 projectors), some were both singing-locked and tutor locked (8/24 neurons, including 1/10 projectors)" (l223)

Did neurons which were both singing and tutor locked show post syllable response to both? Or post syllable for tutor song and pre syllable for singing? Also specific to the similar (copied) syllable? If their model is based on "These ensembles are then re-activated later during 'singing'" (L261), I think that is important point.

Minor comment

Fig5(G) legend: delete one 'showing'

Reviewer #3:

Remarks to the Author:

This paper has been greatly improved by 1) the addition of neural data (significantly more NIf neurons during singing (89 vs. 63 previously) and during tutoring (103 v 49previous); 2) analysis of the timing of NIf activity relative to syllable offsets; 3) clarifying details regarding the model; and 4) simulations to test the combined NIf/HVC model based on recent findings that optogenetic stimulation of NIf with different stimulus parameters can influence syllable duration (Fig. 8). While the simulation does not fully recapitulate the effects of optogenetic stimulation (e.g., sequences of syllables, including stereotyped motifs with 2 or 3 syllables), it was a clever way to test how altering the initial inputs from NIf would affect the formation of HVC networks in the model.

Overall, the revised manuscript provides a significant advance for understanding the mechanisms by which sensory experience can guide motor learning and performance.

Minor comments below are aimed at strengthening the work through clarification:

1) Additional information about the ANOVAs would be helpful. Did the authors perform post-hoc tests to see if the firing rates for particular syllables were significantly different either when the bird was singing or when it was listening to tutor song?

2) In their computational model of NIf, one of the inputs is a syllable onset signal active during tutoring and singing. The signal during singing corresponds to an efference copy of the preparatory motor commands for initiating a syllable. The authors suggest many ways that an efference copy that originates in LMAN could reach NIf or HVC via Uva. Is another possibility that an efference copy could reach Nif or HVC via Area X and A11, a pathway suggested by Hamaguchi and Mooney, 2012?

3) Figure 3E – The utility of the schematic is not clear as there is no mention of how/why the HVC network connectivity would change across developmental stages given that the timing of activity of NIf neurons does not change much (remains clustered prior to syllable onsets at all stages of song learning) across learning stages (Fig. 3C). The NIf/HVC model is more clearly spelled out in the new Figure 8, rendering 3E unnecessary.

4) Remaining concerns about sample size; please check and correct as needed:

i) Pg. 8: 24 were recorded during singing and tutoring; 12/24 were singing-locked; 2/24 were tutor-locked; 8/24 showed changing in firing rate during tutoring and singing. What about the other 2 neurons? Did they not show changes in firing rate during either singing or hearing tutor song?

ii) Methods: 6.2 states that electrophysiological recordings were carried out in 16 juvenile zebra finches (p. 19); 6.3 states that 168 single units were recorded in 20 juvenile birds

iii) Figure 4: Please clarify the n's? Were there a total of 8 NIf HVC neurons (in 8 birds?) that were recorded when birds sang multiple types, and 6/8 (75%) of the cells showed

differential firing for some syllables?

- 5) Figure 7 legend: (B) extra "there
- 6) Figure 8 B,D – missing scale bars?
- 7) No mention of Figure 1B in the text

REVIEWERS' COMMENTS:

Reviewer #1 (Remarks to the Author):

The authors thoroughly addressed all my concerns. I have no further comments.

Reviewer #2 (Remarks to the Author):

In response to my request, providing experimental data for showing Nif instructing HVC neurons, they provided new modeling, Nif-HVC combined modeling based on their Nif recording data and recent publication from the other lab. That did not provide a new conceptual idea, however their Nif recording is importance to test if the modeling concept is reasonable and their experiments and model well confirmed the suggested concept. However, here is my additional comments on the additional parts.

The authors stated “The latencies of individual NifHVC neurons are clustered prior to syllable onsets at all stages of song learning, with 80%, 79% and 83% of bursts occurring prior to syllable onset in subsong, protosyllable and multisyllable stages respectively (Figure 3C). In contrast, the distribution of HVC burst latencies appears to change over development in a way consistent with chains growing from syllable onsets [40]. In adult birds, HVC bursts occur fairly uniformly throughout the song motif [38, 39].” (l164) However, it was not clear why it can be concluded in the following sentence “These results are consistent with a model in which HVC sequences grow from syllable onsets, triggered by inputs from Nif (Figure 3E).” More importantly, it was not stated well how these developmental changes observed in the Nif recording was implemented in their Nif-HVC combined modeling. Did authors want to suggest that connectivity between Nif-HVC alters over development (latency changes could be explained by this?),.

This lack of clarity seems to be similar to that raised by Reviewer 3, related to the diagram in Figure 3. We have updated the diagram and the text (pdf lines 153-160) to clarify this point. We do not suggest that the connectivity between Nif and HVC changes over development, but rather that connectivity within HVC changes over development, as described in Okubo et al., 2015. The key idea is that Nif activates small ensembles of seed neurons in HVC. The resulting reverberant activity in HVC induces the growth of chains in the HVC network, which then supports the propagation of sequences in HVC.

The authors further stated that “This model demonstrates explicitly how the formation of different ensembles in Nif can drive chain splitting in HVC such that different syllables in the song are represented by different chains in HVC” (L357). However, they did not explain how the experimental observation over the development match with the modeling.

The relation between the chain splitting model and the experimental data is a major component of the 2015 Okubo et al paper. We have clarified the text on this point (pdf lines 356-361).

The authors stated “Of the neurons we were able to record 224 during both singing and tutoring, some were exclusively singing-locked (12/24 225 neurons, including 7/10 projectors), some were both singing-locked and tutor locked (8/24 neurons, including 1/10 projectors)” (l223) Did neurons which were both singing and tutor locked show post syllable response to both? Or post syllable for tutor song and pre syllable for singing? Also specific to the similar (copied)

syllable? If their model is based on “These ensembles are then re-activated later during ‘singing’” (L261), I think that is important point.

Of the 8 neurons that were locked at syllable onsets during both singing and tutoring, we observed a range of singing latencies from -6ms to +18ms, with a mean of 9.6ms, and a range of tutoring latencies from +6ms to +45ms, with a mean of +20.3ms. For the one HVC projecting Nlf neuron, the singing latency was -2ms, and the tutoring latency was +26ms. We have added this information to the text.

Regarding the syllable specificity, our aim was to record in young birds, and thus we did not record from the same neurons after the syllable was copied (the recording during singing occurred before the bird copied any syllables).

Minor comment

Fig5(G) legend: delete one ‘showing’

Done.

Reviewer #3 (Remarks to the Author):

This paper has been greatly improved by 1) the addition of neural data (significantly more Nlf neurons during singing (89 vs. 63 previously) and during tutoring (103 v 49previous); 2) analysis of the timing of Nlf activity relative to syllable offsets; 3) clarifying details regarding the model; and 4) simulations to test the combined Nlf/HVC model based on recent findings that optogenetic stimulation of Nlf with different stimulus parameters can influence syllable duration (Fig. 8). While the simulation does not fully recapitulate the effects of optogenetic stimulation (e.g., sequences of syllables, including stereotyped motifs with 2 or 3 syllables), it was a clever way to test how altering the initial inputs from Nlf would affect the formation of HVC networks in the model.

Overall, the revised manuscript provides a significant advance for understanding the mechanisms by which sensory experience can guide motor learning and performance.

Minor comments below are aimed at strengthening the work through clarification:

1) Additional information about the ANOVAs would be helpful. Did the authors perform post-hoc tests to see if the firing rates for particular syllables were significantly different either when the bird was singing or when it was listening to tutor song?

Post-hoc t-tests were performed for the neurons shown in Figures 4 and 6. For both the singing and the tutoring data, the null hypothesis that the neuron fired with the same mean firing rate for the two syllables was rejected ($p < 8.2195e-34$ and $p < 9.4771e-05$ respectively). We have added this information to the text.

2) In their computational model of Nlf, one of the inputs is a syllable onset signal active during tutoring and singing. The signal during singing corresponds to an efference copy of the preparatory motor commands for initiating a syllable. The authors suggest many ways that an efference copy that originates in LMAN could reach Nlf or HVC via Uva. Is another possibility

that an efference copy could reach Nif or HVC via Area X and A11, a pathway suggested by Hamaguchi and Mooney, 2012?

We thank the reviewer for this suggestion, and have added this possibility to the text.

3) Figure 3E – The utility of the schematic is not clear as there is no mention of how/why the HVC network connectivity would change across developmental stages given that the timing of activity of Nlf neurons does not change much (remains clustered prior to syllable onsets at all stages of song learning) across learning stages (Fig. 3C). The Nlf/HVC model is more clearly spelled out in the new Figure 8, rendering 3E unnecessary.

We thank the reviewer for pointing out this lack of clarity. This is similar to Reviewer 2's point about the need for more clarity and context in the presentation of the latency data in Figure 3. This result was a prediction of a previous paper (Okubo et al. 2015). We have added more context and explanation to the text, and adapted the schematic for clarity.

4) Remaining concerns about sample size; please check and correct as needed:

i) Pg. 8: 24 were recorded during singing and tutoring; 12/24 were singing-locked; 2/24 were tutor-locked; 8/24 showed changing in firing rate during tutoring and singing. What about the other 2 neurons? Did they not show changes in firing rate during either singing or hearing tutor song?

Yes, the remaining two neurons did not show changes in firing rate during either singing or tutoring. We have clarified the text.

ii) Methods: 6.2 states that electrophysiological recordings were carried out in 16 juvenile zebra finches (p. 19); 6.3 states that 168 single units were recorded in 20 juvenile birds

Fixed.

iii) Figure 4: Please clarify the n's? Were there a total of 8 Nlf HVC neurons (in 8 birds?) that were recorded when birds sang multiple types, and 6/8 (75%) of the cells showed differential firing for some syllables?

There were a total of 8 Nlf_HVC neurons, from 2 different birds, and 6/8 showed differential firing for some syllables. We have clarified the text.

5) Figure 7 legend: (B) extra "there

Fixed.

6) Figure 8 B,D – missing scale bars?

Scale bars have been added.

7) No mention of Figure 1B in the text

We added a mention of Figure 1B.